# Muscle 4EBP1 activation modifies the structure and function of the neuromuscular junction in mice

Seok-Ting J. Ang[1,2], Elisa M. Crombie [1], Han Dong[1], Kuan-Ting Tan[1], Adriel Hernando[1], Dejie Yu[2,3,4], Stuart Adamson[5], Seonyoung Kim[1], Dominic J. Withers [6,7], Hua Huang[2,3,4] & Shih-Yin Tsai [1,2] ✉

Dysregulation of mTOR complex 1 (mTORC1) activity drives neuromuscular junction (NMJ) structural instability during aging; however, downstream targets mediating this effect have not been elucidated. Here, we investigate the roles of two mTORC1 phosphorylation targets for mRNA translation, ribosome protein S6 kinase 1 (S6K1) and eukaryotic translation initiation factor 4E-binding protein 1 (4EBP1), in regulating NMJ structural instability induced by aging and sustained mTORC1 activation. While myofiber-specific deletion of *S6k1* has no effect on NMJ structural integrity, 4EBP1 activation in murine muscle induces drastic morphological remodeling of the NMJ with enhancement of synaptic transmission. Mechanistically, structural modification of the NMJ is attributed to increased satellite cell activation and enhanced post-synaptic acetylcholine receptor (AChR) turnover upon 4EBP1 activation. Considering that loss of post-synaptic myonuclei and reduced NMJ turnover are features of aging, targeting 4EBP1 activation could induce NMJ renewal by expanding the pool of post-synaptic myonuclei as an alternative intervention to mitigate sarcopenia.

The neuromuscular junction (NMJ) is a chemical synapse between a motor neuron and a skeletal muscle fiber. Optimal neuromuscular transmission is important for muscle contraction and voluntary movement. In the aging population, the decline in skeletal muscle strength typically precedes the loss of muscle mass, and moreover, gaining muscle mass per se is insufficient to promote muscle strength[1]. Such lack of correlation between muscle mass and strength implicates the deterioration of the NMJ in the age-related decline of muscle mass and function[1,2].

Structural instability of the NMJ, which is characterized by the fragmentation of the post-synaptic acetylcholine receptor (AChR) cluster, and is often accompanied by retraction of motor neurons (denervation), has been observed in aged rodents and the mdx mouse, a mouse model of Duchenne muscular dystrophy[2,3]. The mammalian target of rapamycin complex 1 (mTORC1) is a key regulator of cellular metabolism and protein homeostasis[4,5], and it has been recently shown to be involved in the maintenance of NMJ structural integrity[6,7]. Reducing mTORC1 activity in mice with inducible deletion of *Raptor* in the skeletal muscle (iRaptor-mKO) triggers severe fragmentation of post-synaptic AChR clusters and enhances the expression of denervation marker neural cell adhesion molecule (NCAM)[8]. Likewise, enhancing mTORC1 activity by conditional deletion of its negative

[1]Department of Physiology, Yong Loo Lin School of Medicine, National University of Singapore, Singapore 117593, Singapore. [2]Healthy Longevity Translational Research Programme, Yong Loo Lin School of Medicine, National University of Singapore, Singapore 117456, Singapore. [3]Electrophysiology Core Facility, Yong Loo Lin School of Medicine, National University of Singapore, Singapore 117544, Singapore. [4]Cardiovascular Diseases Program, National University of Singapore, Singapore 117599, Singapore. [5]Buck Institute for Research on Aging, Novato, CA, USA. [6]Metabolic Signalling Group, Medical Research Council Clinical Council London Institute of Medical Sciences (LMS), Du Cane Road, London W12 0NN, UK. [7]Institute of Clinical Sciences (ICS), Faculty of Medicine, Imperial College London, Du Cane Road, London W12 0NN, UK. ✉e-mail: phsts@nus.edu.sg

regulator, tuberous sclerosis complex 1 (*Tsc1*), in the skeletal muscle (TSC1mKO) also induces fragmentation of the AChR cluster and impairs denervation-induced AChR turnover, a critical process for muscle/NMJ repair and regeneration in response to injury[7]. On the other hand, partially suppressing mTORC1 activity with rapamycin preserves the stability of AChR clustering and muscle strength in natural aging and TSC1mKO mice[6]. Taken together, these studies suggest that the tight regulation of mTORC1 is important for the maintenance of NMJ structural integrity. However, the factors downstream of mTORC1 in regulating NMJ structural integrity remain unclear.

The ribosomal protein S6 kinase 1 (S6K1) and eukaryotic translation initiation factor 4E (eIF4E)-binding protein 1 (4EBP1) are two of the most well-characterized mTORC1 downstream targets regulating mRNA translation. Upon activation of the mTORC1 signaling cascade, mTORC1 phosphorylates S6K1 leading to its activation and subsequent phosphorylation of S6, as well as other components of the translation machinery. Whereas phosphorylation of 4EBP1 by mTORC1 releases its binding from eIF4E, freeing this initiation factor to promote cap-dependent translation. Reducing mTORC1-S6K1 or mTORC1-4EBP1 signaling has been reported to maintain muscle health in different contexts. For example, muscle-specific deletion of *S6k1* improves muscle function and extends the lifespan of a mouse model with A-type lamin-related muscular dystrophy[9], while activation of 4EBP1 signaling in the skeletal muscle protects it from age- and obesity-induced metabolic decline[10]. The beneficial effects of chronic rapamycin treatment are mediated, in part, through inhibition of mTORC1-induced phosphorylation of S6K1 and 4EBP1[11].

In the present study, we examine whether reducing mTORC1-S6K1 or mTORC1-4EBP1 signaling could improve the structural stability of the NMJ. We show that while *S6k1* inactivation does not protect NMJ structural integrity in aging muscle, nor in the background of enhanced mTORC1 activity, transgenic activation of 4EBP1 in myofibers triggers dramatic remodeling of NMJ with an increase in post-synaptic myonuclei. Structural modification of the NMJ upon 4EBP1 activation is accompanied by increased satellite cell activation and enhanced post-synaptic AChR turnover. Our result reveals that mTORC1-4EBP1 signaling is critical for NMJ function.

## Results

### NMJ alterations during aging

To examine age-dependent remodeling of the NMJ, we performed immunofluorescence staining on quadriceps muscles of 6- and 28-month-old male and female control mice (wildtype mice). The pre-synaptic structures were stained with neurofilament and synaptophysin, marking the axon and synaptic vesicles, respectively. The post-synaptic structure was stained with α-bungarotoxin (BTX), which labeled the AChR, and DAPI was used to label the nuclei in the muscle sections. Typically, NMJs of young mice have a continuous pretzel-like post-synaptic structure with corresponding apposed pre-synaptic components, while during aging, NMJ is fragmented into disconnected, discrete AChR clusters (Fig. 1a). Quantitative analysis of the NMJ was conducted by characterizing typical morphological features of age-induced NMJ alterations that include (1) fragmentation of post-synaptic structures, assessed by counting the number of discontinuous AChR fragments, (2) reduction in the number of post-synaptic myonuclei (transcriptionally specialized myonuclei clustered beneath the post-synaptic membrane supporting NMJ function), and (3) the loss of innervation, measured by the degree of overlap between pre- (neurofilament and synaptophysin) and post-synaptic (BTX) components, and reported here as the Mander's colocalization coefficient score[2,12–14]. Alternatively, staining of NCAM was used as a biomarker for myofiber denervation. NCAM is a glycoprotein highly expressed at the NMJ (junctional NCAM), which accumulates intracellularly in the skeletal muscle

upon denervation and is thus a commonly used marker of myofiber denervation[15].

Our analysis of age-associated NMJ properties was consistent with published data, but only in male mice (Fig. 1a–d). In male mice, we observed an increase in the number disconnected post-synaptic AChR fragments (Fig. 1b), a reduction in the number of post-synaptic myonuclei (Fig. 1c), and a decrease in Mander's colocalization coefficient, indicating a loss of innervation (Fig. 1d). In female mice, however, we did not observe any fragmentation of post-synaptic structure nor loss of innervation, even though the age-induced loss of post-synaptic myonuclei was still observed. Our observations suggest a sex-specific effect of aging on NMJ structural modification.

Consistent with reduced innervation of NMJ measured by the overlap of pre- and post-synaptic components (Fig. 1d and Supplementary Fig. 1a), we found a significant increase in the percentage of myofibers with intracellular accumulation of NCAM during aging (Fig. 1e, f), suggesting that the denervation response was enhanced in a subset of myofibers. After embryogenesis, intracellular accumulation of NCAM was previously observed in denervated adult myofibers or disease-associated degenerating–regenerating myopathy[16]. Since the regeneration capacity is halted in aging muscle, our data suggested that the denervation response is enhanced, which leads to muscle degeneration in a subset of myofibers. Further analysis on NCAM staining revealed that the mean cross-sectional area (CSA) of NCAM-positive fibers was significantly smaller than NCAM-negative fibers in young mice, while the mean CSA of NCAM-positive fibers in old mice was no different from NCAM-negative fibers (Fig. 1f). The observation suggests that enhanced denervation response marked by intracellular accumulation of NCAM in old animals is associated with an age-induced muscle degeneration. Moreover, the increased number of NCAM-positive fibers was also accompanied with induction of another denervation marker, the fetal AChRγ subunit gene (*Chrng*) expression, in aged mice (Supplementary Fig. 1b)[6,13]. Note that the expression of other genes reported upon acute denervation were varied in aging muscles which is reflective of the heterogeneity of the aging process.

During aging, increased phosphorylation of the ribosomal protein S6 (pS6) is typically used as a measurement of enhanced mTORC1-S6K1 activation. At 6 months of age, the pS6 signal was localized to the NMJ and the myonuclei (Fig. 2a) and rarely accumulated in the myofiber. However, an increased intracellular accumulation of pS6 was detected in aged mice (Fig. 2a, b), in line with enhanced mTORC1 activity in previous findings[6,17]. Compared to the percentage of fibers with intracellular accumulation of NCAM (Fig. 1f), there were more fibers with intracellular accumulation of pS6 (Fig. 2b) in 28-month-old mice. The angular shaped fibers with intracellular pS6 staining were phenotypically similar to those found with intracellular NCAM accumulation observed in aged mice (Fig. 2c).

### Chronic activation of mTORC1 drives sarcopenia-like NMJ remodeling

Given the similar morphology of pS6- and NCAM-positive fibers, it is likely that pS6-positive fibers are denervated, suggesting that mTORC1 activation could be a cause or consequence of a damaged myofiber, thereby marking the denervated myofiber before NCAM. To evaluate the cause-and-effect relationship of chronic mTORC1 activation, we generated a myofiber-specific knockout of TSC1 in the mouse by crossing *Tsc1^{f/f}* mice with *Ckmm-cre* mice (referred to as TSC1mKO). Consistent with the previous reports[17,18], we found that deletion of *Tsc1* causes hyper-phosphorylation of S6 and 4EBP1 in skeletal muscle (Supplementary Fig. 1c). In addition, we found that adult TSC1mKO mice had comparable percentages of pS6-positive fibers versus those of naturally aged mice (Fig. 2). The size disparity between pS6-positive and pS6-negative fibers was also abolished in TSC1mKO mice, similar to that observed in naturally aged mice (Fig. 2b). Yet, unlike naturally aged mice versus young mice, the CSA

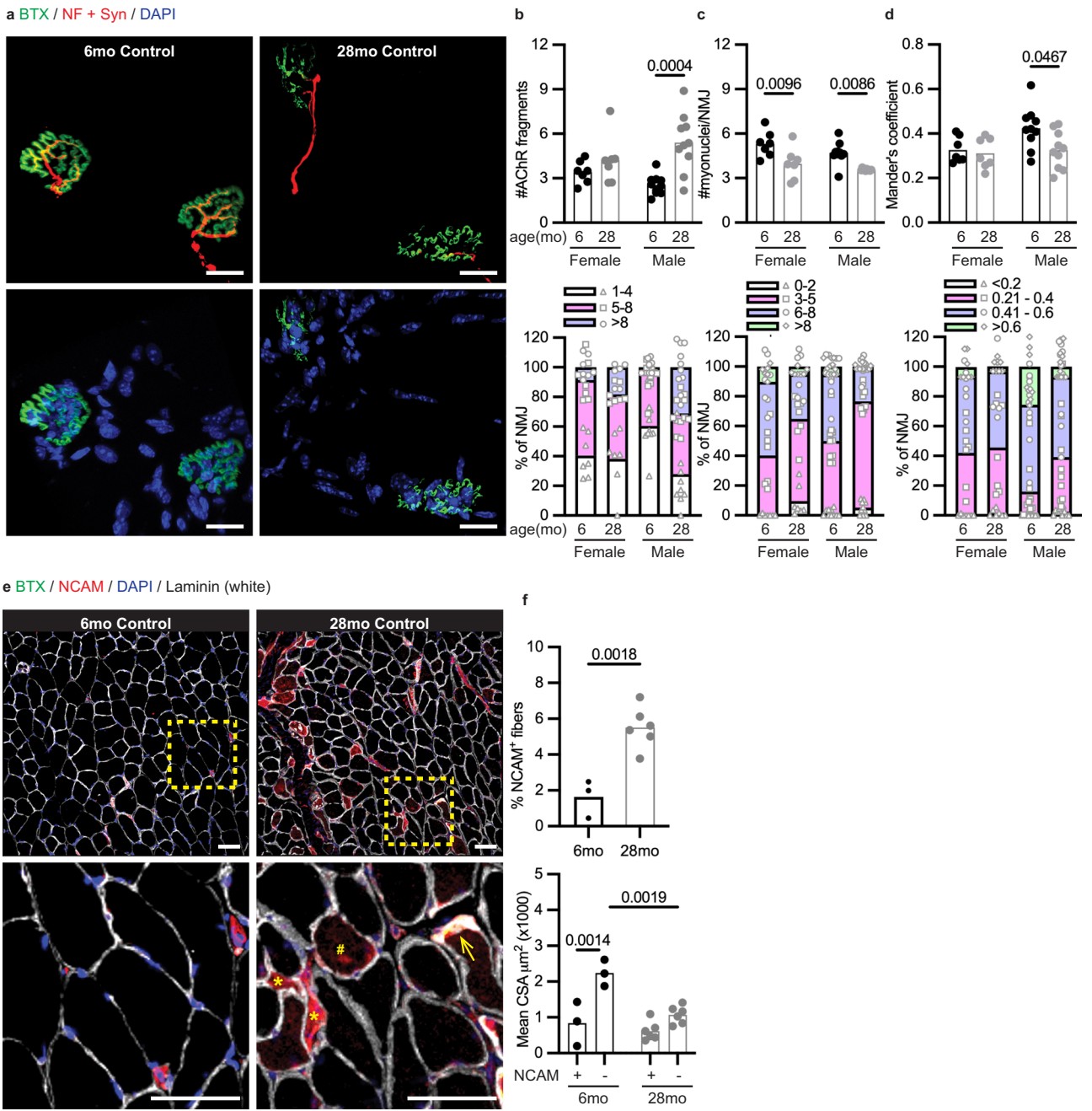

**Fig. 1 | Sexual dimorphism of NMJ remodeling during aging. a** Representative confocal images of NMJs from longitudinal sections of the quadricep muscle of 6- and 28-month-old male control mice (6mo Control and 28mo Control, *n* = 10 per group). Top panels = BTX (green) and neurofilament (NF; red) + synaptophysin (Syn; red); Bottom panels = BTX (green) and DAPI (blue). Scale bar = 20 μm. **b**–**d** Quantification of NMJ morphological properties of male and female control mice, which include (**b**) the number of AChR fragments, (**c**) the number of post-synaptic myonuclei, and (**d**) the overlap between pre- and post-synaptic components as measured by the Mander's colocalization coefficient. The top figures are mean values per animal and the bottom figures are percentage distribution for each parameter. Sample size: 6mo Control Female and 28mo Control Female, *n* = 7 per group; 6mo Control Male and 28mo Control Male, *n* = 10 per group; on average, 25 NMJs were analyzed per mouse. **e** Representative confocal images of BTX (green),

NCAM (red), DAPI (blue) and Laminin (white) staining from cross-sections of the gastrocnemius muscle of male control mice (6mo Control, *n* = 3; 28mo Control, *n* = 6). The insets are shown in the bottom panels. Open arrows = junctional NCAM; Asterisks = intracellular NCAM in angular, odd-shaped fibers; Hash = intracellular NCAM accumulation in abnormally enlarged fibers. Scale bars = 50 μm. **f** Quantification of the percentage of NCAM-positive myofibers (top panel) and mean CSA of NCAM-positive (NCAM+) and NCAM-negative (NCAM-) fibers (bottom panel). Sample size: 6mo Control Male, *n* = 3; 28mo Control Male, *n* = 6; on average, 800 myofibers were analyzed per mouse. Statistical significance: NMJ quantification (Fig. 1b–d) and myofiber analysis of NCAM staining (Fig. 1f, bottom panel), two-way ANOVA followed by Tukey's post-hoc pairwise comparison; Myofiber analysis-NCAM-positive myofibers (Fig. 1f, top panel), two-tailed unpaired student *t*-test. Only a *P* value of less than 0.05 is labeled in the figure.

of pS6-positive fibers from TSC1mKO mice (*P* = 0.006; Fig. 2b) was larger than that of control mice. Next, we analyzed the NMJs of these mice at 12 months old to determine age-related abnormalities. This timepoint was chosen because TSC1mKO mice showed advanced myopathy[17,18].

At 12 months of age, a robust increase in fragmentation of post-synaptic AChR clusters was seen in the NMJs of TSC1mKO mice compared to age-matched littermate controls (Fig. 3a, c and Supplementary Fig. 1d). The AChR cluster fragmentation was more pronounced in 12-month-old TSC1mKO versus 28-month-old control mice. In the

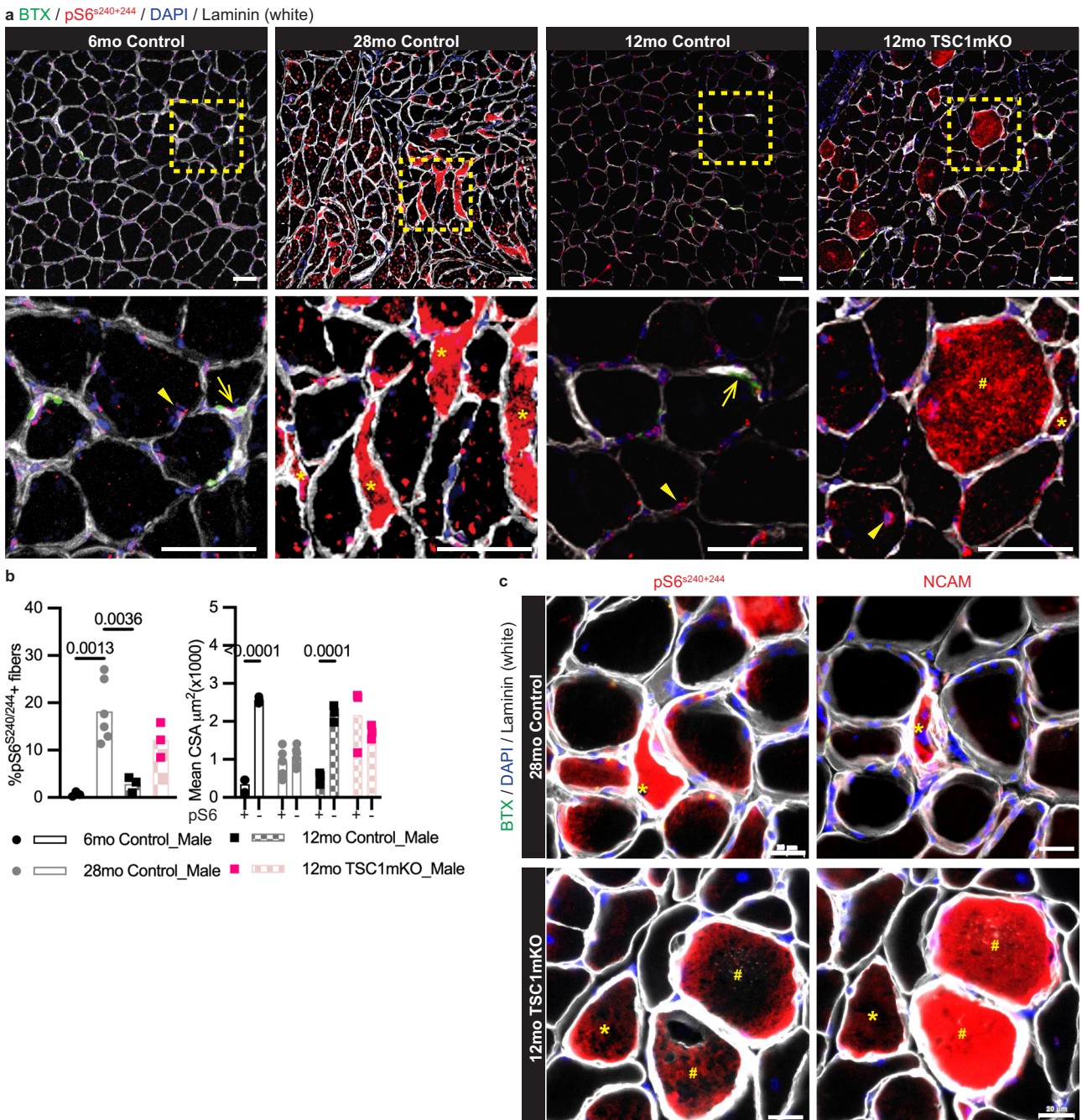

**Fig. 2 | Chronic activation of mTORC1 in aging muscle. a** Representative confocal images of BTX (green), pS6$^{S240+244}$ (red), DAPI (blue) and Laminin (white) staining from cross-sections of the gastrocnemius muscle of 6- and 28-month-old male control mice and 12-month-old control and TSC1mKO male mice (6mo Control, $n=3$; 28mo Control, $n=6$; 12mo Control, $n=3$; 12mo TSC1mKO, $n=3$). The insets are shown in the bottom panels. Asterisks = intracellular pS6$^{S240+244}$ in angular, odd-shaped fibers; Hash = intracellular pS6$^{S240+244}$ accumulation in abnormally enlarged fibers; Open arrow = junctional pS6$^{S240+244}$; Arrowhead = pS6$^{S240+244}$ in the nucleus, including central nucleus. Scale bars = 50 μm. **b** Quantification of the percentage of pS6$^{S240+244}$-positive myofibers (left panel; statistical significance was determined by one-way ANOVA followed by Tukey's post-hoc pairwise comparison is used for statistical analysis) and mean CSA of pS6$^{S240+244}$ -positive (pS6+) and pS6$^{S240+244}$ -negative (pS6-) fibers (right panel; statistical significance was determined by two-way ANOVA followed by Tukey's post-hoc pairwise comparison is used for statistical analysis). Sample size: 6mo Control Male, $n=3$; 28mo Control Male, $n=6$; 12mo Control Male, n = 3; 12mo TSC1mKO Male, $n=3$; on average, 800 myofibers were analyzed per mouse. Only a $P$ value of less than 0.05 is labeled in the figure. **c** Representative images of serial sections stained with BTX (green), DAPI (blue), Laminin (white), and pS6$^{S240+244}$ (red, left panels) or NCAM (red, right panels) from cross-sections of the gastrocnemius muscle of 28-month-old control male (top panels; $n=6$) and 12-month-old TSC1mKO male (bottom panels; $n=3$). Asterisks = intracellular pS6$^{S240+244}$ and NCAM in angular, odd-shaped fibers; Hash = intracellular pS6$^{S240+244}$ and NCAM accumulation in abnormally enlarged fibers. Scale bars = 20 μm.

context of post-synaptic myonuclei, no difference was observed between 12-month-old control and TSC1mKO mice. Likewise, the NMJs of 12 and 28-month-old mice had similar level of post-synaptic myonuclei (Fig. 3d), suggesting that post-synaptic myonuclei number had

already declined in the NMJs by 12 months of age, which might explain the lack of effect in TSC1mKO mice. Similar phenomena were also observed in Mander's colocalization coefficient (Fig. 3e). We also noted that the subtle difference in NMJ fragmentation between male and

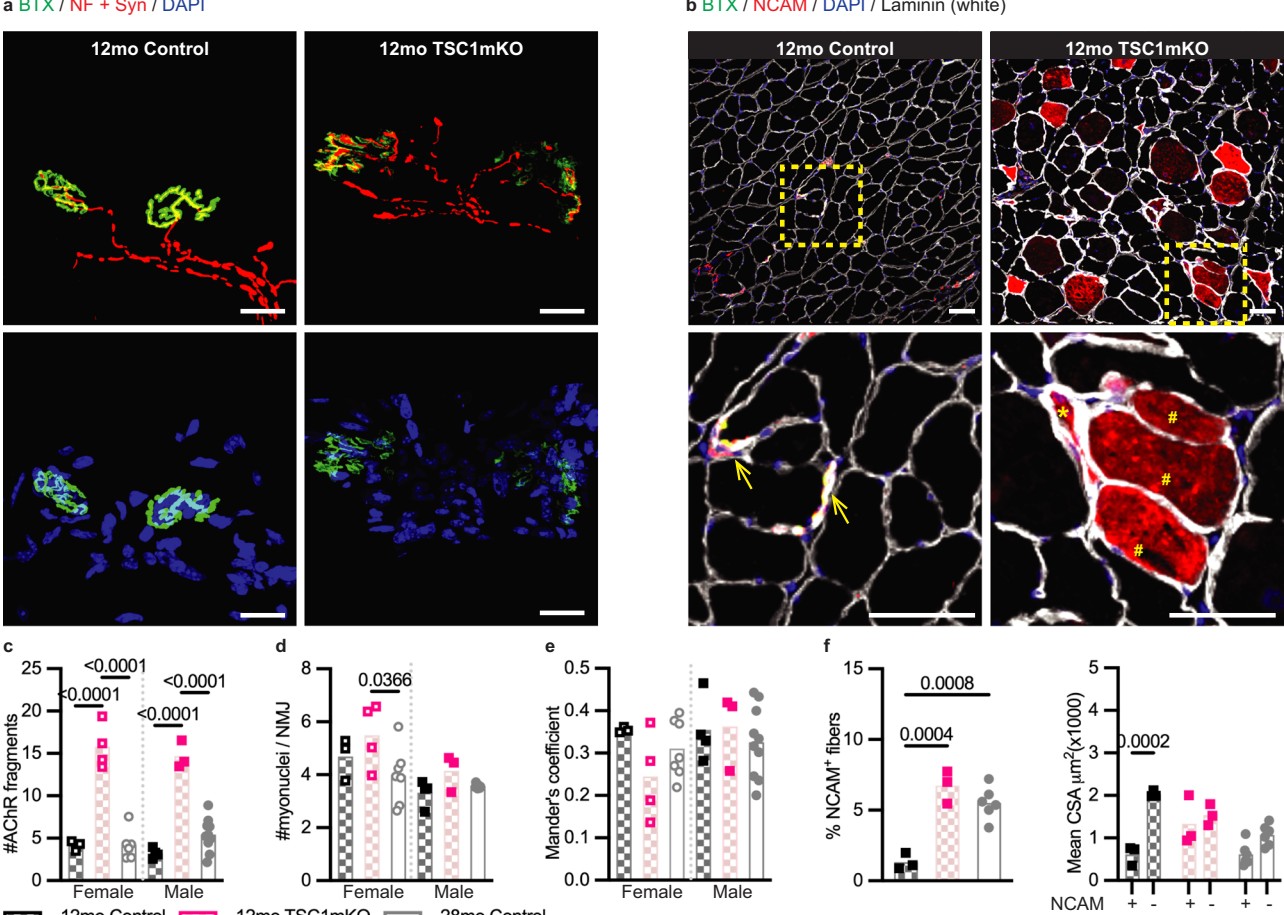

**Fig. 3 | Chronic activation of mTORC1 drives sarcopenia-like NMJ remodeling.** **a** Representative confocal images of NMJs from longitudinal sections of the quadriceps muscle of 12-month-old male mice (Control, n = 4; TSC1mKO, n = 3). Top panels = BTX (green) and neurofilament (NF; red) + synaptophysin (Syn; red); Bottom panels = BTX (green) and DAPI (blue). Scale bar = 20 µm. **b** Representative confocal images of BTX (green), NCAM (red), DAPI (blue) and Laminin (white) staining from cross-sections of the gastrocnemius muscle of 12-month-old male mice (Control, n = 3; TSC1mKO, n = 3). The insets are shown in the bottom panels. Open arrows = junctional NCAM; Asterisks = intracellular NCAM in angular, odd-shaped fibers; Hash = intracellular NCAM accumulation in abnormally enlarged fibers. Scale bars = 50 µm. **c–e** Quantification of NMJ morphological properties that include (**c**) the number of AChR fragmentation, (**d**) the number of post-synaptic myonuclei, and (**e**) Mander's colocalization coefficient. Sample size: 12mo Control Female, n = 3; 12mo TSC1mKO Female, n = 4; 28mo Control Female, n = 7; 12mo Control Male, n = 4; 12mo TSC1mKO Male, n = 3; 28mo Control Male, n = 10; on average, 25 NMJs were analyzed per mouse. **f** Quantification of the percentage of NCAM-positive myofibers (left panel) and mean CSA of NCAM-positive (NCAM + ) and NCAM-negative (NCAM-) fibers (right panel). Sample size: 12mo Control Male, n = 3; 12mo TSC1mKO Male, n = 3; 28mo Control Male, n = 6; on average, 800 myofibers were analyzed per mouse. Statistical significance: NMJ quantification (Fig. 3c–e) and myofiber analysis of NCAM staining (Fig. 3f, right panel), two-way ANOVA followed by Tukey's post-hoc pairwise comparison; Myofiber analysis-NCAM-positive myofibers (Fig. 3f, left panel), one-way ANOVA followed by Tukey's post-hoc pairwise comparison. Only a P value of less than 0.05 is labeled in the figure.

female aged NMJs were not observed in the TSC1mKO mouse model, suggesting that mTORC1 hyperactivation could accelerate and drive NMJs fragmentation (Supplementary Fig. 1d). The percentage of NCAM-positive fibers was increased (Fig. 3b, f) and correlated with the upregulation of the AChRγ subunit gene (*Chrng*) and the other denervation-induced atrophy genes such as *Runx1*[19], and *Gadd45a*[20] in TSC1mKO mice (Supplementary Fig. 1e). Adult TSC1mKO mice had comparable percentages of NCAM-positive myofibers versus those of naturally aged mice. The size disparity between NCAM-positive and NCAM-negative myofibers was also abolished in TSC1mKO mice, reassembling that observed in naturally aged mice (Fig. 3f). Similar to pS6-positive fibers (Fig. 2b), the CSA of NCAM-positive myofibers (P = 0.0749; Fig. 3f) from TSC1mKO mice was larger than their age-matched control mice (Fig. 2c). These observations imply that activation of mTORC1 is the driver for muscle degeneration and denervation. When mTORC1 hyperactivation can not be repressed, the myofiber eventually undergoes denervation-induced degeneration. Consistently, muscle functions evaluated by four-limb hanging tests in vivo

showed a significant reduction in TSC1mKO (Supplementary Fig. 6c). Our findings, therefore, suggest that TSC1mKO mice may serve as a preclinical model to capture the pre-symptomatic stages of sarcopenia developed over a shorter period of time.

### Genetic deletion of *S6k1* has no effect on NMJ remodeling

Pharmacological inhibition of mTORC1 activity by rapamycin treatment has been shown to improve age-induced muscle atrophy and NMJ instability[6]. S6K1 and 4EBP1 are two canonical downstream targets of mTORC1 and represent two distinct pathways through which rapamycin may potentially act. Therefore, we determined whether mTORC1 signaling through either S6K1 or 4EBP1 affected age- or mTORC1-induced NMJ structural alterations. To test our hypothesis, we generated two mouse lines – (1) myofiber-specific deletion of *S6k1*, by crossing *S6k1*^f/f mice with mice expressing *Ckmm-cre* (referred to as S6K1mKO; Supplementary Fig. 3d), and (2) double knockout of *Tsc1* and *S6k1* in the myofiber, by crossing *S6k1*^f/f;*Tsc1*^f/f mice with *S6k1*^f/f; *Tsc1*^f/f;*Ckmm-cre* mice to delete the S6K1 gene in the background of

sustained mTORC1 activation (referred to as S6K1-TSC1mKO; Supplementary Fig. 5h) – to study the effect of genetic deletion of *S6k1* on age- and mTORC1 hyperactivation-induced NMJ remodeling, respectively. S6K1 has previously been proposed as a dominant S6K isoform to regulate growth in skeletal muscle independent of S6K2[21], and whole-body knockout of *S6k1* female mice had better muscle performance with age and was long-lived[22]. Additionally, muscle-specific deletion of *S6k1* is sufficient to extend lifespan in *Lmna* knockout mice, a genetic model for the study of laminopathy, toward a similar degree of rapamycin treatment[9]. Yet, we found surprisingly, muscle-specific deletion of *S6k1* has no effect on skeletal muscle growth nor maintenance with age. S6K1mKO mice have comparable muscle type distribution and size to control mice (Supplementary Fig. 2a–f). In concordance with muscle size, S6K1mKO mice have comparable muscle function to their control littermates (Supplementary Fig. 2g). Muscle-specific deletion of *S6k1* also did not rescue muscle function and myopathy such as inclusions, degenerated basophilic fibers and basophilic "ragged" fibers in TSC1mKO mouse background (Supplementary Fig. 6).

The persistence of weak S6 phosphorylation in young S6K1mKO mice and the increased intensity of S6 phosphorylation in the S6K1-TSC1mKO mice (Supplementary Fig. 3d) suggest that other cellular mechanisms might have compensated for the absence of S6K1 in the skeletal muscle[23,24], despite western blot analysis showed the lack of S6K1 protein, demonstrating successful deletion of *S6k1*. Indeed, it has been previously shown that upregulation of S6K2 compensates for the deletion of *S6k1* in vivo[23]. Even though we did not observe an increase in total protein expression of S6K2, the phosphorylation of Ser423 at S6K2 by MEK/ERK signaling, which initiates S6K2 activation[25], was slightly up-regulated in S6K1mKO mouse muscle (Supplementary Fig. 3e), suggesting that S6K2 might have compensated for S6 phosphorylation in the absence of S6K1. The unchanged pS6 level was also observed in muscle-specific deletion of *S6k1* in the *Lmna* knockout mouse background, which extends lifespan[9].

Next, we performed morphological analyses of the NMJs of young and old S6K1mKO mice versus their age-matched littermate controls. Even though pS6 signal might not be a good marker for S6K1 activities, we did find that the localization of the pS6 signal in the myonuclei was largely preserved to location of the NMJ and the myonuclei, and the intracellular pS6 accumulation was reduced in aged S6K1mKO mice (Supplementary Fig. 3a–c), unlike naturally aged mice (Fig. 2a). Yet, deletion of *S6k1* in the background of TSC1mKO (S6K1-TSC1mKO) did not affect the intracellular accumulation of pS6 (Supplementary Fig. 3b, c).

The number of AChR fragments was not significantly different between 6- and 28-month-old male S6K1mKO NMJ (Supplementary Fig. 4b) and the number of post-synaptic myonuclei was preserved in 28-month-old female S6K1mKO NMJ (Supplementary Fig. 4c). Regarding Mander's colocalization coefficient, no significant difference was observed between 6- and 28-month-old NMJ in male S6K1mKO mice (Supplementary Fig. 4d) but increased in NCAM-positive myofibers with age persisted (Supplementary Fig. 4e, f). In general, the deletion of *S6k1* has a subtle effect on NMJ integrity likely due to heterogenity of the aging process. Similarly, deletion of *S6k1* in the background of TSC1mKO had no protective effect on NMJ integrity (Supplementary Fig. 4) and on denervation-associated gene expression (Supplementary Fig. 5f, g). Collectively, our data indicate that the deletion of *S6k1* has no significant effect on NMJ instability during aging nor in the background of overactive mTORC1.

### Transgenic activation of 4EBP1 in the skeletal muscle induces drastic morphological changes at the NMJ

Another direct target of mTORC1 that mediates protein synthesis is 4EBP1. mTORC1-dependent phosphorylation of 4EBP1 at Thr37/46 releases its inhibitory interaction with the translation initiation factor eIF4E to facilitate CAP-dependent translation[26]. To assess the potential

contribution of 4EBP1 to NMJ structural remodeling, we first analyzed the subcellular expression of *Eif4ebp1* mRNA in extensor digtorum longus muscle using RNAscope in situ hybridization. We observed strong expression of *Eif4ebp1* mRNA that was enriched around post-synaptic myonuclei, which also co-expressed the synaptic gene C*hrne* that encodes for the adult AChRε subunit (Fig. 4a). Our RNAscope analysis is also consistent with published RNAseq analysis from NMJ and non-NMJ regions that showed two-fold increase of *Eif4ebp1* in the NMJ regions[6].

We had previously generated a mouse line with transgenic expression of a mutant form of *Eif4ebp1* within mTOR phosphorylation site (threonine to alanine at amino acid positions 37 and 46) to yield a constitutively active form of the protein (referred to as 4EBP1mt mice) that is non-responsive to mTORC1 regulation[10]. The 4EBP1mt transgene expression is repressed by a *loxP*-flanked *Stop* codon cassette and is induced in the skeletal muscle by crossing 4EBP1mt mice with mice expressing *Ckmm-cre* (referred to as 4EBP1mt-muscle mice). 4EBP1mt-muscle mice develop normally but have reduced growth in muscle size with increased slow-twitch myofibers and mitochondrial activities. Even though muscle atrophy was observed in young 4EBP1mt-muscle mice, those mice were more active and were protected from aging-induced muscle dystrophy resembling the effect of calorie restriction on skeletal muscle[10]. Given the localization of *Eif4ebp1* at the NMJ and the preservation of muscle function and metabolism in aged 4EBP1mt-muscle mice, we next examined those mice to evaluate the effect of loss of function of mTORC1-4EBP1 signaling on age-induced NMJ remodeling. In addition, we induced the transgene expression of 4EBP1mt in the background TSC1mKO (referred to as 4EBP1mt-TSC1mKO) to investigate the effect of 4EBP1 mutation on mTORC1-induced NMJ remodeling (Supplementary Fig. 5h).

Surprisingly, we observed a very drastic remodeling in the NMJs of 4EBP1mt-muscle mice. Transgenic 4EBP1 activation in the skeletal muscle induced severe fragmentation of the NMJs (Fig. 4d), from a pretzel-like structure into highly branched grape-like aggregates (Fig. 4b) that expanded to occupy a larger endplate area (Supplementary Fig. 5a). The fragmentation induced by 4EBP1 mutant protein expression in the skeletal muscle yielded more fragments than aged mice (Supplementary Fig. 5c). In conjunction with increased fragmentation, 4EBP1mt-muscle NMJs had a significantly higher number of post-synaptic myonuclei (Fig. 4e and Supplementary Fig. 5d) and increased gene expression encoded AChR subunit genes (Supplementary Fig. 5e) as compared to age-matched controls. Moreover, 4EBP1mt-muscle mice have reduced of Mander's colocalization coefficient (Fig. 4f) and increased frequency of NCAM-positive fibers (Fig. 4c, g). Interestingly, the level of fragmentation, the number of post-synaptic myonuclei, and the proportion of NCAM-positive myofibers were also not different between young and old 4EBP1mt-muscle mouse NMJs (Fig. 4d–f). These observations indicate that NMJs of 4EBP1mt-muscle mice are fragmented at a young age but remained relatively stable and did not undergo further structural remodeling during aging. The 4EBP1mt-muscle NMJ phenotype partially resembles that of iRaptor-mKO mice, in which post-synaptic AChR cluster fragmentation and increased NCAM-positive myofibers were also reported[8], suggesting that 4EBP1 is a critical downstream factor mediating mTORC1 activities during NMJ remodeling.

In the context of 4EBP1-TSC1mKO mice, myofiber-specific expression of 4EBP1mt in the background of high mTORC1 activity drives the phenotype of NMJ remodeling (Fig. 4b). Similar to 4EBP1mt-muscle mice, 4EBP1-TSC1mKO mice had a higher degree of AChR fragmentation (Fig. 4d and Supplementary Fig. 5c), and number of post-synaptic myonuclei (Fig. 4e and Supplementary Fig. 5d) indicative of more fragmented NMJs with a larger myonuclei domain. Surprisingly, the proportion of NCAM-positive fibers (Fig. 4c, g) and *Gadd45a* expression (Supplementary Fig. 5g, h) in 4EBP1mt-TSC1mKO mice was lower than that of TSC1mKO mice, though it was still higher than that

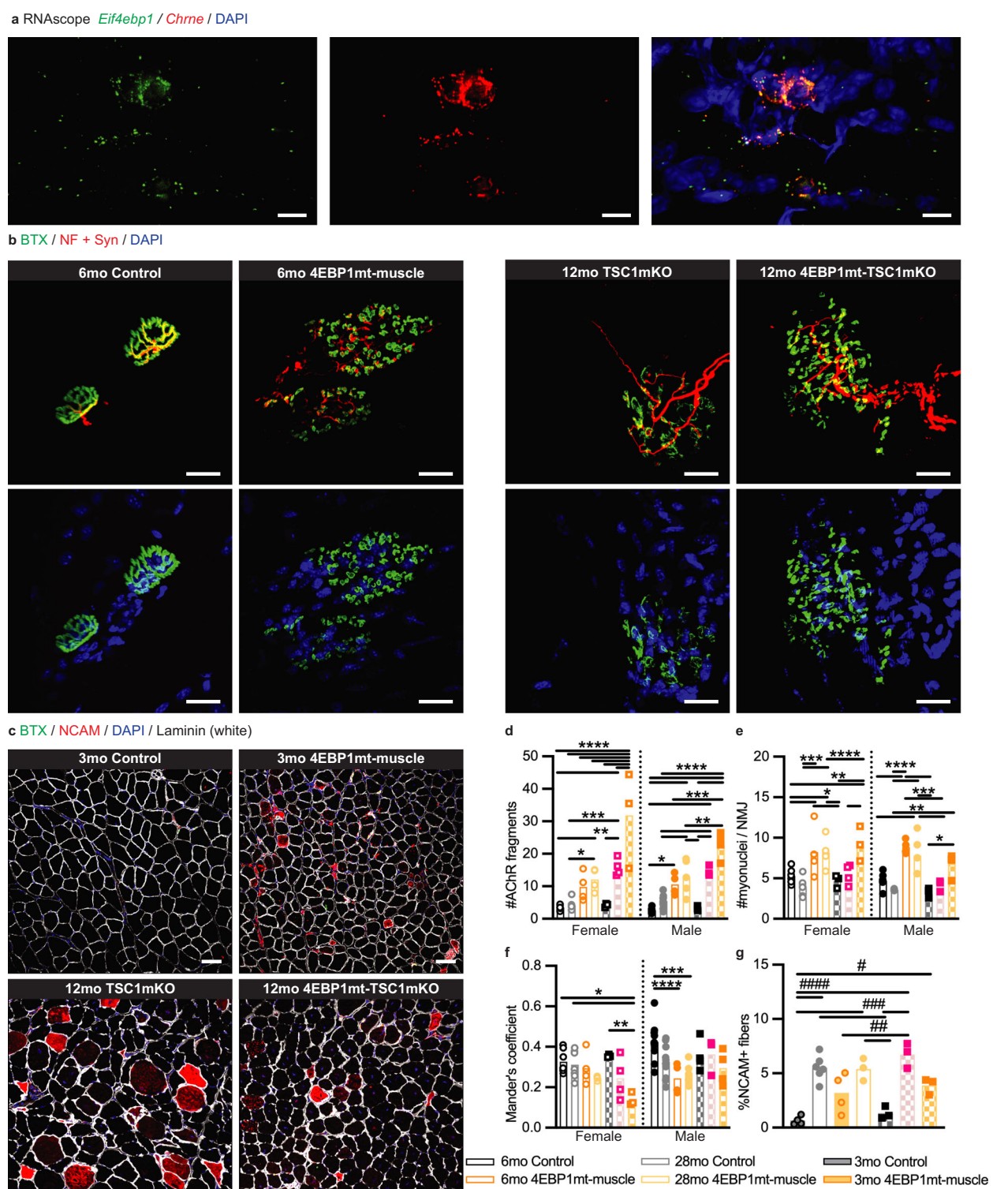

of age-matched control mice. These observations collectively imply that the denervation response is reduced with 4EBP1 activation in the background of mTORC1 hyperactivity, similar to rapamycin treatment in these mice[6]. Activation of 4EBP1 has also reduced the frequency of inclusions and degenerated basophilic fibers present in TSC1mKO mouse background as well as muscle function is improved in 4EBP1mt-TSC1mKO mice (Supplementary Fig. 6). Together, our results indicate that 4EBP1 mediates mTORC1 activities in regulating NMJ structure and function, where mTORC1-4EBP1 signaling is tightly regulated to preserve NMJs. Complete inhibition of mTORC1 signaling by deletion of

*raptor* or activation of 4EBP1causes dramatic NMJ remodeling, whereas 4EBP1 activation represses mTORC1 overactivity- mediated denervation.

## Structural aberrations in 4EBP1mt-muscle NMJs result in enhanced NMJ synaptic transmission

Since intracellular NCAM was increased in young 4EBP1mt-muscle mice, which suggested an active process of denervation or ongoing cycles of myofiber degeneration and regeneration, we next addressed whether the structural aberrations observed at the NMJs of 4EBP1mt-

**Fig. 4 | Skeletal muscle-specific 4EBP1 activation induces severe structural alterations of the NMJ. a** Representative confocal images showing RNAscope in situ hybridization of *Eif4ebp1*(green) and *Chrne* (red) mRNA counterstained with DAPI (blue) in longitudinal sections of the extensor digitorum longus muscle from 3-month-old control male mice (*n* = 3). Scale bar = 20 μm. **b** Representative confocal images of NMJs from longitudinal sections of the quadriceps muscle from male mice (6mo Control, *n* = 10; 6mo 4EBP1mt-muscle, *n* = 4; 12mo TSC1mKO, *n* = 3; 4EBP1mt-TSC1mKO Male, *n* = 4). Top panels = BTX (green) and neurofilament (NF; red) + synaptophysin (Syn; red); Bottom panels = BTX (green) and DAPI (blue). Scale bar = 20 μm **c** Representative confocal images of BTX (green), NCAM (red), DAPI (blue) and Laminin (white) staining of cross-sections of the gastrocnemius muscle from male mice (3mo Control and 4EBP1mt-muscle, *n* = 4 per group; 12mo TSC1mKO, and 4EBP1mt-TSC1mKO, *n* = 3 per group). Scale bar = 50 μm. **d**–**g** Quantification of NMJ morphological properties which include (**d**) the number of AChR fragmentation, (**e**) the number of post-synaptic myonuclei, (**f**) Mander's colocalization coefficient, and (**g**) quantification of the percentages of NCAM-

positive myofibers. Sample size: NMJ quantification (Fig. 4d–f), 6mo and 28mo Control Female, *n* = 7 per group; 6mo and 28mo 4EBP1mt-muscle Female, *n* = 4 per group; 12mo Control and 4EBP1mt-TSC1mKO Female, *n* = 3 per group; 12mo TSC1mKO Female, *n* = 4 per group; 6mo and 28mo Control Male, *n* = 10 per group; 6mo 4EBP1mt-muscle Male, *n* = 4; 28mo 4EBP1mt-muscle Male, *n* = 5; 12mo Control Male and 4EBP1mt-TSC1mKO Male, *n* = 4 per group; 12mo TSC1mKO Male, *n* = 3 per group; on average, 25 NMJs were analyzed per animal. Myofiber analysis (Fig. 4g), 3mo Control and 4EBP1mt-muscle Male, *n* = 4 per group; 28mo Control Male, *n* = 6; 28mo 4EBP1mt-muscle Male, *n* = 3; 12mo Control, TSC1mKO, and 4EBP1mt-TSC1mKO Male, *n* = 3 per group; on average, 800 myofibers were analyzed per mouse. Statistical significance: *$P < 0.05$, **$P < 0.01$, ***$P < 0.001$, ****$P < 0.0001$, two-way ANOVA followed by Tukey's post-hoc pairwise comparison; #$P < 0.05$, ##$P < 0.01$, ###$P < 0.001$ ####$P < 0.0001$ one-way ANOVA followed by Tukey post-hoc pairwise comparison. Only a *P* value of less than 0.05 would be labeled in the figure. Detailed statistical analysis of Fig. 4d–g are listed in Supplementary Tables 4–7, respectively.

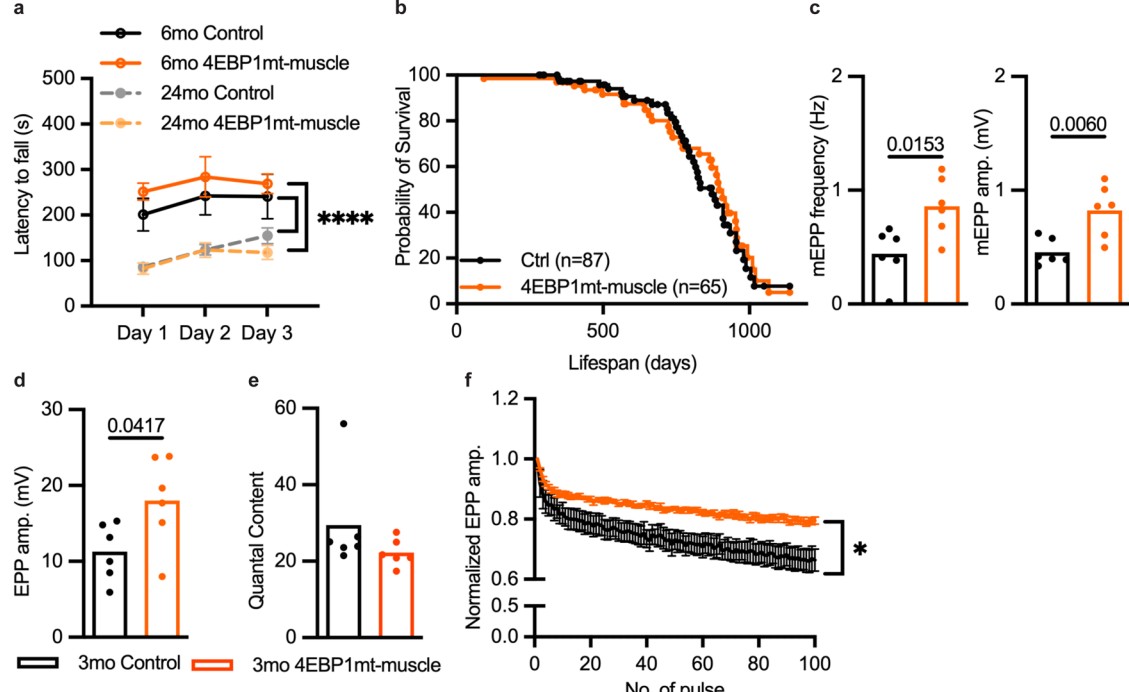

**Fig. 5 | 4EBP1 activation enhances neuromuscular synaptic transmission.**
**a** Motor coordination performance as measured by the rotarod in 6- and 24-month-old male mice over a 3-day period (6mo Control and 4EBP1mt-muscle, *n* = 5 per group; 24mo Control, *n* = 9; 24mo 4EBP1mt-muscle, *n* = 7. Data are presented as mean values ± SEM. ****$P < 0.0001$ and statistical significance was determined by two-way ANOVA followed by Tukey's post-hoc pairwise comparison of 6- versus 24-month-old mice within indicated genotype. **b** Lifespan of male 4EBP1mt-muscle mice versus their littermate controls (Control Male, *n* = 87, 4EBP1mt-muscle Male, *n* = 65). The comparison of survival rates between groups was examined by Log-rank (Mantel–Cox) test. The median lifespans of the control and 4EBP1mt-muscle

mice were 872 days and 897 days, respectively. **c**–**e** Average mEPP frequency (left) and mEPP amplitude (right) (**c**), EPP amplitude (**d**) and quantal content (**e**) recorded from the NMJs of 3-month-old male mice. Statistical significance was determined by two-tailed unpaired student *t*-test. Only a *P* value of less than 0.05 would be labeled in the figure. **f** Normalized EPP amplitude of 3-month-old male mice in response to repeated nerve stimulation at 1 Hz. Data are presented as mean values ± SEM. *$P < 0.05$; statistical significance was determined by two-way ANOVA followed by Tukey's post-hoc pairwise comparison of genotype effect. Sample size: NMJ neurotransmission (Fig. 5c–f), 3mo Control and 4EBP1mt-muscle Male, *n* = 6 per group; on average, 8 myofibers were recorded per animal.

muscle mice affected neuromuscular function. Motor coordination performance evaluated by the rotarod assay was similar between control and 4EBP1mt-muscle mice at young and old age. Both genotypes showed a comparable age-related decline in motor coordination (Fig. 5a). Moreover, 4EBP1mt-muscle mice had comparable lifespan to control mice (Fig. 5b) and we did not observe any muscle weakness nor paralysis (data not shown), which might account for the severe NMJ morphology changes in 4EBP1mt-muscle mice.

Electrophysiological recording from individual muscle fibers in the diaphragm was used to directly assess synaptic transmission of the NMJs from 3-month-old mice, in which dramatic fragmentation is

already present in 4EBP1mt-muscle mice (Supplementary Fig. 7a). We first recorded and analyzed the frequency and amplitude of miniature endplate potentials (mEPPs), which are events of local depolarizing potentials elicited by the spontaneous release of acetylcholine (ACh). The increase in both frequency and amplitude of mEPP was observed in 4EBP1mt-muscle mice (Fig. 5c and Supplementary Fig. 7b) indicating an increase in pre-synaptic release probability and post-synaptic sensitivity to quantal release, respectively. Similarly, we also found enhancement in response to nerve stimulation in the NMJs of 4EBP1mt-muscle mice. Thus, the amplitude of EPPs was significantly elevated in 4EBP1mt-muscle mice versus controls (Fig. 5d and Supplementary

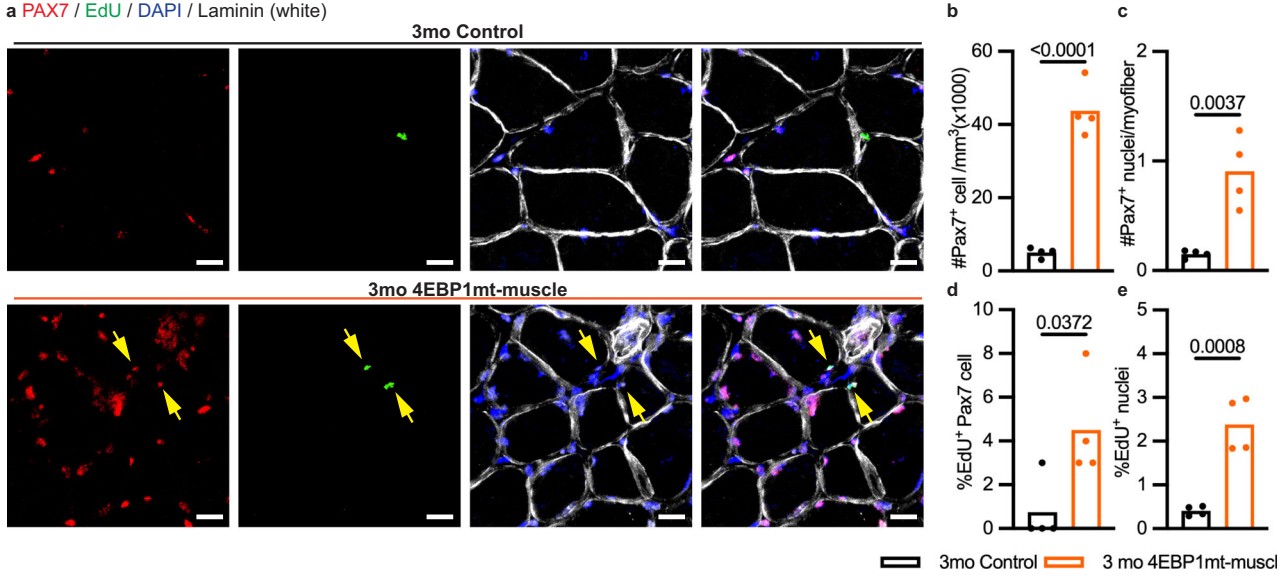

**a** PAX7 / EdU / DAPI / Laminin (white)

**Fig. 6 | 4EBP1 activation increases satellite cell number and activity.**
**a** Representative confocal images of PAX7 (red), EdU (green), DAPI (blue) and Laminin (white) staining of cross-sections of the gastrocnemius muscle from 3-month-old male mice (3mo Control and 4EBP1mt-muscle, $n = 4$ per group). Solid Arrows = EdU and PAX7 double-positive satellite cells. Scale bar = 20 μm.
**b**, **c** Quantification of the number of Pax7-positive nuclei normalized to (**b**) area and

(**c**) number of myofibers. **d**, **e** Quantification of the percentages of (**d**) EdU and PAX7 double-positive cell and (**e**) EdU-positive nuclei. Sample size: PAX7 and EdU staining in cross section (Fig. 6b–e), 3mo Control and 4EBP1mt-muscle Male, $n = 4$ per group; on average, 400 myofibers sampled per animal. Statistical significance: PAX7 and EdU staining in cross section (Fig. 6b–e), Two-tailed unpaired student $t$-test. Only a $P$ value of less than 0.05 would be labeled in the figure.

Fig. 7b). The quantal content (i.e., EPP/mEPP, which quantifies the number of ACh vesicles released in response to a single nerve stimulation) was not different between the two groups (Fig. 5e). This suggests that the post-synaptic AChRs of 4EBP1mt-muscle were more sensitive to the release of neurotransmitters. Indeed, the AChRs of 4EBP1mt-muscle NMJs were highly fragmented and dispersed over a larger endplate area (Supplementary Fig. 5a), which might have resulted in a higher magnitude of post-synaptic membrane depolarization. However, it is worth noting that the area occupied by AChR was not different between control and 4EBP1mt-muscle NMJs (Supplementary Fig. 5b). Next, we performed repeated stimulation on the phrenic nerve to evaluate neurotransmission fatigue in the diaphragm. With 100 pulses given at 1 Hz, the EPPs recorded in the NMJs of 4EBP1mt-muscle and control mice were suppressed in relation to the first pulse. The magnitude of suppression was similar in the first 10 pulses for both groups, but from the 11th pulse onwards, the suppression of EPP was attenuated in 4EBP1mt-muscle NMJs (Fig. 5f; $P < 0.05$). This indicates that the NMJs of 4EBP1mt-muscle mice had better neurotransmission recovery to repeated stimulation. When stimulated at a higher frequency of 10 Hz, both 4EBP1mt-muscle and control NMJs showed a similar level and pattern of EPP suppression (Supplementary Fig. 7c). In all, we conclude that despite severe structural remodeling of NMJ that reminisces NMJ instability, neurotransmission function of 4EBP1mt-muscle NMJs is not compromised.

**Transgenic activation of 4EBP1 induces muscle regeneration.** Lastly, we examine the potential cause or mechanism that might contribute to the morphological instability of NMJ in 4EBP1mt-muscle mice without compromising its physiological neurotransmission function. Consistent with previously reported in young 4EBP1mt-muscle mice[10], we found that activation of 4EBP1 resulted in a higher proportion of central nucleated fibers in aged 4EBP1mt-muscle mice (Supplementary Fig. 7d) and in the background of TSC1mKO (Supplementary Fig. 7e). The presence of a higher number of central nucleated fiber and post-synaptic nuclei led us to posit that myofiber-specific activation of 4EBP1 might have active myofiber regeneration as

a result of higher number and activity of satellite cells, the stem cell pool in the skeletal muscle. To test our hypothesis, we performed immunofluorescence staining of PAX7, a commonly used marker of satellite cell. At 3 months of age, we observed more PAX7-positive cells in 4EBP1mt-muscle mice versus controls (Fig. 6a–c). In addition, we determined the proliferative activity of the satellite cells by performing a 3-day EdU pulse. We found that the number of EdU and PAX7 double-positive cells was higher in the 4EBP1mt-muscle group than age-matched controls (Fig. 6a, d), thereby confirming that more satellite cells had proliferated with 4EBP1 activation in the myofibers. We also found that 4EBP1mt-muscle mice had higher number and percentage of EdU-positive myonuclei (Fig. 6e). The majority of EdU-positive nuclei were found at the extra-synaptic sites, but a small fraction of post-synaptic myonuclei associated with BTX was found to be EdU-positive (Fig. 7a, c). Moreover, a higher proportion of NMJs with EdU-positive nuclei were detected in 4EBP1mt-muscle mice as compared to age-matched littermate controls (Fig. 7d). Taken together, the observations suggest that under myofiber-specific activation of 4EBP1, satellite cell number and activity were increased. The integration of newly synthesized myonuclei underneath the NMJ accounts for the expansion in the post-synaptic myonuclear domain in 4EBP1mt-muscle NMJs.

The extensive remodeling seen in 4EBP1mt-muscle and 4EBP1mt-TSC1mKO mice (specifically, fragmentation of AChR cluster and expansion of endplate area) suggests a high rate of AChR turnover. To confirm the notion in 4EBP1mt-muscle mice, we performed the AChR turnover assay, where two distinct pools of AChRs were labeled by intra-muscular injection of BTX conjugated to different fluorophores. First, BTX conjugated to Alexa Fluor-647 (BTX-AF647) was administered into the tibialis anterior to label pre-existing AChRs. Ten days later, the same muscle was injected with BTX conjugated to Alexa Fluor-488 (BTX-AF488). The mice were euthanized 24 h after, and the muscle was dissected and sectioned for imaging to reveal the relative distribution of BTX-AF647 ("old receptors") and BTX-AF488 ("new receptors") signal intensities. In contrast to controls with low AChR turnover, 4EBP1mt-muscle mice showed a relatively high AChR

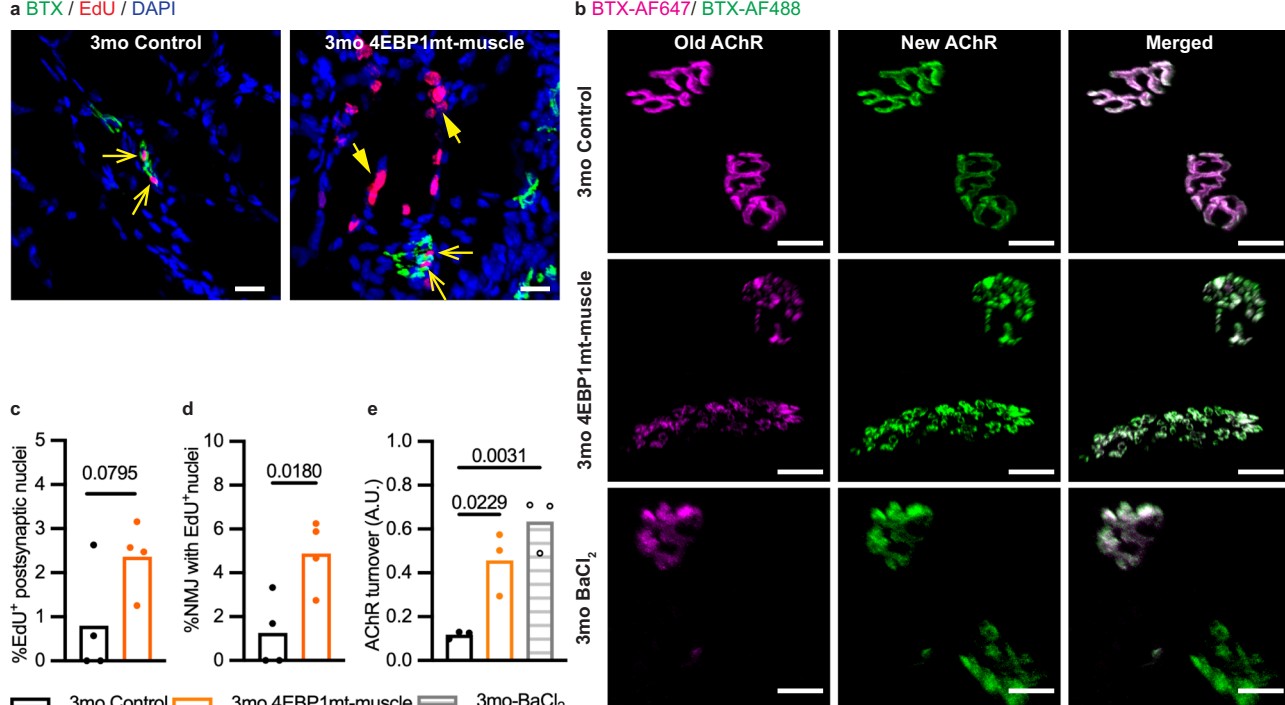

**Fig. 7 | 4EBP1 activation increases the rate of AChR turnover. a** Representative confocal images of BTX (green), EdU (red) and DAPI (blue) staining of longitudinal sections of the quadriceps muscle from 3-month-old male mice (3mo Control and 4EBP1mt-muscle, $n = 4$ per group). Open arrows = EdU-positive post-synaptic myonuclei; Solid arrows = EdU-positive non-synaptic myonuclei. Scale bar = 50 μm. **b** Representative confocal images showing 'Old AChR' labeled with BTX-AF647 (magenta) and 'New AChR' with BTX-AF488 (green) from longitudinal sections of the tibialis anterior muscle of 3-month-old Control ($n = 3$) and 4EBP1mt-muscle ($n = 3$) male mice as well as control mice treated with BaCl$_2$ to induce muscle damage (3mo BaCl$_2$, $n = 3$). Scale bar = 20 μm. **c, d** Quantification of the percentage of (**c**) EdU-positive post-synaptic nuclei and (**d**) NMJ with EdU-positive nuclei. Sample size: 3mo Control and 4EBP1mt-muscle Male, $n = 4$ per group; on average, 15,000 myonuclei were sampled per animal. Statistical significance was determined by two-tailed unpaired student $t$-test. **e** Quantification of AChR turnover in the NMJs. Sample size: 3mo Control, 4EBP1mt-muscle, Control treated with BaCl$_2$ Male, $n = 3$ per group, on average 20 NMJs were sampled per animal. Statistical significance was determined by one-way ANOVA followed by Tukey post-hoc pairwise.

turnover (Fig. 7b, e). The level of AChR turnover in 4EBP1mt-muscle mice was similar to that of barium chloride-induced muscle damage (Fig. 7e), thereby confirming that AChR turnover is accelerated in NMJs of 4EBP1mt-muscle mice akin to those experiencing muscle damage. Overall, these data indicate that muscle regeneration and NMJ turnover were enhanced, which both attribute to inducing drastic morphological changes and enhancing neurotransmission at the NMJ of 4EBP1mt-muscle mouse.

## Discussion

As the interface between the motor neuron and muscle fiber, optimal synaptic transmission at the NMJ is required to coordinate muscle contraction and maintain muscle mass. NMJ structural integrity is paramount to this process, and its alterations are suggested to play a pivotal role in the sarcopenia-related decline of muscle mass and function. In the present study, we show that downstream targets of mTORC1, S6K1, and 4EBP1, have differential effects on NMJ structural integrity. While mTORC1-S6K1 and mTORC1-4EBP1 activities and expression are enriched at NMJ and post-synaptic myonuclei, inactivation of S6K1 had no effect on NMJ structure at a young or old age, nor in the background of enhanced mTORC1 activity. In contrast, transgenic activation of 4EBP1 in myofibers triggered dramatic remodeling of NMJ into fragmented grape-like aggregates with increased post-synaptic myonuclear domain. Intriguingly, while this structural phenotype is typically associated with reduced NMJ function, we reveal that motor coordination was unaffected in 4EBP1mt-muscle mice. Their NMJs also had enhanced synaptic transmission functions, accompanied by enhanced muscle and NMJ regenerative capacity.

The activities and expression of two canonical downstream targets of mTORC1, S6K1/S6 and 4EBP1, are found to be enriched at the NMJ and they differentially contribute to its structural integrity. Though *S6k1* was successfully deleted, the level of S6 phosphorylation persisted. It is possible that hyperactivity of mTORC1 caused by *Tsc1* deletion led to an overdrive of the mTORC1-S6K-S6 axis by activating S6K2 or RSK. Indeed, it has been shown that *Rps6kb2* (encoding S6K2) and *Rps6ka1* (encoding RSK1) were shown to be highly enriched in the NMJ region compared to the non-NMJ region, whereas *Rps6kb1* (encoding S6K1) has the opposite expression pattern[6].

In the context of 4EBP1mt-muscle mice, the loss of function of mTORC1-4EBP1 signaling induced severe NMJ remodeling. Myofiber-specific expression of 4EBP1mt induced an NMJ phenotype that partially resembles that of iRaptor-mKO mice, in which the key component of mTORC1 is deleted at adulthood, marked by increased fragmentation and increased NCAM accumulation in myofibers[8]. This is expected as iRaptor-mKO also enhances 4EBP1 activity. Yet, the increase in the number of post-synaptic myonuclei and activated muscle regeneration observed in 4EBP1mt-muscle was not reported in iRaptor-mKO mice[8,27]. Moreover, the underlying mechanisms driving NMJ remodeling in these two mouse lines are different. The structure instability of iRaptor-mKO NMJ is attributed to impairment in autophagy flux and mitochondrial dysfunction[8], though muscle mass and function are unaffected[27]. However, in the case of 4EBP1mt-muscle mice, the 4EBP1 activation-induced NMJ phenotype is an effect of enhanced muscle/NMJ regeneration, where skeletal muscle of 4EBP1mt-muscle shows atrophy with enhanced mitochondrial function and basal autophagy[10]. Detailed elucidation of mTORC1

downstream effector interactions in regulating NMJ structure integrity and functionality will be required for better therapeutic interventions targeting mTORC1 to treat sarcopenia.

Fragmentation of post-synaptic AChR cluster is a hallmark of the NMJ during aging[14,28], occurs in response to denervation, and is observed in mouse model of muscle dystrophy such as the mdx mouse[3]. This phenomenon has sometimes been interpreted as NMJ degeneration, and intervention such as caloric restriction or restoration of autophagy activity reduced NMJ fragmentation concomitantly with improved muscle function with age[14,29]. Yet, in aged diaphragm muscle, no correlation was found between fragmentation and neuromuscular transmission[30]. The mice with hyperactive mTORC1 earlier in the embryogenesis have only shown subtle signs of defective neuromuscular transmission in limb muscle, even though severe fragmentation and denervation of NMJ were reported[6]. Whereas in mdx mice, fragmentation was associated with reduced mEPP amplitude, increased quantal content with no change to EPP, indicative of homeostatic pre-synaptic compensatory response to the reduced postsynaptic sensitivity for ACh[3,31]. However, we could not rule out that the severe denervation phenotypes observed in mdx mice might be attributed to undefined defects in the motor neurons since mdx mice carry genetic mutation of dystrophin, whose expression is not restricted to the skeletal muscle. Further investigation is required to distinguish the cause-and-effect relationship between AChR fragmentation and denervation. It is possible that denervation triggers the degeneration/regeneration of the muscle fibers, which in turn causes AChR fragmentation and that AChR fragmentation is a compensatory mechanism to preserve neuromuscular transmission in the physiological setting. In the present study, the genetic modification of 4EBP1 was restricted to fully differentiated myofibers. The fragmentation phenotype driven by chronic 4EBP1 activation is likely a result of cycles of degeneration and regeneration of the myofibers and NMJs. Upon muscle damage or denervation, the existing AChR cluster disassembles rapidly and is replaced with new receptors that become fragmented, in conjunction with the appearance of central nuclei, a sign of regeneration[32,33]. High incidences of central nucleation have been reported in myofibers of 4EBP1mt-muscle mice[10], and the percentage of fibers containing central nuclei is higher in 4EBP1mt-TSC1mKO versus TSC1mKO mice. Muscle regeneration is responsible for repairing damaged skeletal muscle, which requires activating PAX7-expressing satellite cells to produce myogenic progenitors and subsequently fusing with myofibers. Consequently, we found a higher density and enhanced cell proliferation activity of PAX7-positive cells in 4EBP1mt-muscle mice, consistent with enhanced regeneration capacity. In addition, the number of newly added post-synaptic nuclei was significantly increased in 4EBP1mt-muscle mice, indicative of NMJ renewal. The addition of new myonuclei to post-synaptic sites also likely explains the increase in the post-synaptic myonuclear domain in 4EBP1mt-muscle mouse NMJ. Concurrent with the activation of satellite cells, genes encoded for AChR subunits were upregulated to support increased AChR turnover, another feature of muscle/NMJ renewal. Another striking observation is that we found the enrichment of 4EBP1 expression in post-synaptic myonuclei of wildtype mice and transgenic activation of 4EBP1 drastically increase the number of post-synaptic myonuclei. Whether 4EBP1 is required for myonuclear functional commitment to post-synaptic myonuclei in muscle would be interesting for further investigation. Since the loss of post-synaptic myonuclei is the feature of aging, targeting 4EBP1 activation might be also beneficial to NMJ renewal by expanding the pool of postsynaptic myonuclei.

Despite severe fragmentation of AChR, neurotransmission in 4EBP1mt-muscle mice was not compromised. We observed increased mEPP and nerve stimulation-evoked EPP amplitude without alteration to quantal content. These observations suggest that the enhancement of synaptic transmission in the NMJs with 4EBP1 activation is likely contributed by post-synaptic changes that rendered the muscle more sensitive to neurotransmitter release. For instance, increased post-synaptic AChR concentration and/or AChR's affinity for ACh can lead to larger electrophysiological events. Hence, we found upregulation of AChR genes in 4EBP1mt-muscle mice and expansion of endplate area, which could imply a higher number of AChR receptors. Another possible explanation for enhanced neurotransmission in 4EBP1mt-muscle mice is due to changes in fiber type composition. The electrophysiological profiles of fast-twitch glycolytic and slow-twitch oxidative fibers are different. For example, fast-twitch EDL and tibialis anterior have lower mEPP or mEPC than slow-twitch soleus[6,34]. Thus, fiber type transformation to an oxidative phenotype might explain the enhanced neuromuscular transmission in 4EBP1mt-muscle mice[10]. In the same vein, 4EBP1mt-muscle NMJs demonstrated improved transmission fatigability to repeated nerve stimulations that are likely due to such fiber type transformation.

In conclusion, our findings demonstrate that mTORC1 downstream effectors, S6K1 and 4EBP1, play distinct roles in mediating NMJ maintenance and remodeling. While S6K1 has a non-obligatory role in NMJ maintenance, 4EBP1 is a critical player in maintaining NMJ structural integrity. Therefore, reducing mTORC1-4EBP1 signaling induces high regeneration of muscle and NMJ associated with enhanced neuromuscular synaptic transmission.

## Methods
### Mouse models
Five transgenic mouse strains were used in this study: TSC1mKO (*Tsc1^{f/f}; Ckmm-cre*), S6K1mKO (*S6k1^{f/f}; Ckmm-cre*), S6K1-TSC1mKO (*Tsc1^{f/f}; S6k1^{f/f}; Ckmm-cre*), 4EBP1mt-muscle (*4EBP1mt; Ckmm-cre*) and 4EBP1mt-TSC1mKO (*Tsc1^{f/f}; 4EBP1mt; Ckmm-cre*). *Tsc1^{f/f}* and *Ckmm-Cre* mice were obtained from The Jackson Laboratory, *S6k1^{f/f}* mice[35] were provided by Dominic J. Withers and 4EBP1mt mice were previously generated in our laboratory. All the mice were bred into C57BL/6 background for 5 generations. Control mice were littermates that did not express the Cre-recombinase. Mice were maintained on a 12-hr light/dark cycle with free access to water and standard mouse chow (Teklad global 18% protein rodent diets, 2018, Evigo). Mice will be monitored closely for general health concerns. While developmental problems are unlikely in these animals, at 9 months of age, spine problems (kyphosis) and muscle weakness are expected in TSC1mKO mice. In this case, food pellet and water gel will be placed in the cage should animals have problems assessing the food hopper and water gel.

For NMJ morphological and muscle biochemistry analysis, S6K1mKO, 4EBP1mt-muscle, TSC1mKO, S6K1-TSC1mKO, 4EBP1mt-TSC1mKO mice and their corresponding littermate controls were aged to the respective timepoints indicated in the Results section. Typically, S6K1mKO and 4EBP1mt-muscle mice in the 6-month-old group were euthanized between 6 and 7 months of age, and those in the 28-month-old group were euthanized between 26 and 28 months of age. TSC1mKO, S6K1-TSC1mKO, 4EBP1mt-TSC1mKO were euthanized between 12 and 13 months of age. In addition, one group of 4EBP1mt-muscle and control mice used for AChR turnover, EdU assays RNAScope in situ hybridization, and electrophysiological study were euthanized between 2 and 3 months of age. Mice were euthanised by isoflurane inhalation and skeletal muscle tissue were dissected and harvested. All hind limb muscles used in this study were dissected and frozen in liquid nitrogen-cooled isopentane.

### Immunofluorescence staining, imaging, and structural analysis of the neuromuscular junction (NMJ)
The quadriceps muscle was sectioned in the longitudinal plane at 40 µm. The free-floating method was used for NMJ staining. The cryosections were first fixed in 4% paraformaldehyde (PFA, MP Biomedicals, 02150146) for 10 min, permeabilized in 0.5% triton-X100

(Sigma, T8787) in phosphate buffered saline (PBS, Gibco, 18912014) for 15 min, and then blocked in 5% bovine serum albumin (BSA, MP Biomedicals, 0882045) in 0.5% Tween-20 (Sigma, P9416) in 1X PBS for 4 h. After blocking, the sections were incubated at room temperature overnight on an orbital shaker with the primary antibody cocktail consisting of rabbit anti-neurofilament (1:1500; Cell Signaling, 2837) and rabbit anti-synaptophysin (1:300, Invitrogen PA1-1043) to stain for the presynaptic compartment. After washing, the sections were then incubated with the corresponding donkey anti-rabbit IgG Alexa Fluor (AF) 564 (1:1000; Invitrogen, A10042), α-BTX (1:2000; Invitrogen, B13422), and DAPI (1 μg/ml; Roche, 10236276001) for 2 h at room temperature on an orbital shaker in the dark. The sections were mounted on Poly-lysine coated slides (Fisher Scientific, J2800AMNZ), dried slightly, and cover-slipped with Prolong Gold Antifade (Invitrogen, P36930) mounting medium. All images were acquired using an Olympus confocal microscope (FV1000 or FV3000, National University of Singapore School of Medicine Confocal Microscopy Unit) at 60X objective with an oil immersion lens. Image analysis was performed using Imaris Software (9.5). First, three-dimensional (3D) surfaces of α-BTX staining—i.e., the post-synaptic apparatus—were created for quantification of the number of disconnected objects (number of AChR fragments). Subsequently, the 3D surfaces generated from the post-synaptic staining were used as a mask to overlay on the synaptophysin and neurofilament (pre-synaptic) channel. The "masked" pre-synaptic staining was then used for colocalization analysis using the colocalization tool. The extent of innervation was determined by the degree of overlap between the pre- and post-synaptic regions and expressed as Mander's colocalization coefficient. Next, 3D surfaces of the DAPI channel were created to count for the number of post-synaptic myonuclei per NMJ. Only nuclei on the post-synaptic surface, i.e., nuclei on the concave surface of BTX 3D reconstruction and those with more than 70% of the volume within the BTX surface, were considered post-synaptic myonuclei. For motor endplate analysis, the BTX channel of individual NMJ was rotated to the en-face orientation with Imaris, and the Z-stack was projected to a 2D image. Fiji software was then used to outline the NMJ to calculate the endplate area and the AChR area, i.e., area of BTX signal. On average, approximately 25 NMJs were sampled and analyzed per muscle.

### Immunofluorescence staining and analysis for NCAM and pS6$^{S240+244}$

The gastrocnemius muscle was sectioned transversely at 10–15 μm and attached to poly-lysine coated slides. After drying in room temperature, the sections were fixed with 2% PFA, permeabilized in 0.3% Triton-X100 in PBS and blocked in 10% goat serum (Gibco, 16210072) + 5% BSA in 0.1% Tween-20. Primary antibodies were incubated overnight at room temperature in a humidified chamber. The following antibodies were used for immunofluorescence: rabbit anti-Phospho-S6$^{S240+244}$ (1:100, Cell Signaling, 2215), rabbit-anti NCAM (1:100, Millipore Merck, AB5032), rat anti-laminin (1:300, Abcam, ab11576). Secondary antibody cocktails consisting of anti-rabbit IgG AF-594 (1:500, Invitrogen, A11012) and anti-rat IgG AF-647 (1:500, Invitrogen, A21247) as well as BTX-AF488 and DAPI were incubated for 1 h at room temperature. After staining was completed, the sections were coverslipped in Prolong Gold Antifade mounting medium. Muscle fibers were imaged randomly at 20X using the FV3000 Olympus Confocal microscope. No signal was observed with the omission of primary antibodies, which served as a negative staining control. On average, 800 fibers were sampled per muscle section for each set of immunostaining.

Quantification was done automatically, using Myosoft, a freely available macro for Fiji. The macro was developed to quantify the CSA of different muscle fiber types based on their intracellular staining. Since our objective was to quantify the number and CSA of myofibers that were intracellularly stained with NCAM and pS6, we had utilized Myosoft to automate our data analysis to prevent biases. Briefly,

Myosoft used a pre-trained machine-learning algorithm to delineate myofiber boundaries using the channel image of laminin staining. The segmented image was then used to generate a region of interest (ROI) that was subsequently overlaid on channel image of NCAM or pS6 staining to extract myofiber positive for NCAM or pS6 based on intensity within the ROIs.

### Immunofluorescence staining, imaging, and analysis for PAX7

The gastrocnemius muscle was sectioned transversely at 10μm and attached to poly-lysine coated slides. After air-drying, the sections were fixed in 4% PFA, followed by antigen retrieval with Target Retrieval Solution (Citrate pH 6.1, Dako) for 10 min at 95 °C. The sections were then permeabilized in 0.5% TritonX100 in PBS for 15 min, followed by blocking with AffiniPure Fab Fragment Goat Anti-Mouse IgG (H + L) (1:10; Jackson Immuno Research Laboratories, AB_2338476) diluted in 5% BSA in 0.1% Tween-20 in PBS for 1 h. Primary antibody cocktail comprising mouse IgG1-anti-PAX7 (1:50; Developmental Studies Hybridoma Bank) and rat anti-laminin (1:300) was incubated at room temperature overnight, and then respective secondary antibodies (goat-anti-mouse IgG-AF568 [Invitrogen, A-11031] and goat anti-rat IgG-AF-647, 1:500 for both) with DAPI for 1 h. After staining was completed, the sections were coverslipped in Prolong Gold Antifade mounting medium. Muscle fibers were imaged at 40X objective. On average, 400 myofibers were sampled randomly per muscle. The number of PAX7 cells was calculated using Imaris software using the same method quantifying the number of post-synaptic myonuclei of the NMJs.

### Cell proliferation assay and analysis

EdU (5-ethynyl-2'deoxyuridine, Lumiprobe, 20540) was injected intraperitoneally (50 mg/kg) for 3 days into 2-month-old 4EBP1mt-muscle mice. The animals were euthanized 28 days after the first dose. For visualization of EdU-incorporated nuclei, the quadriceps muscle was sectioned longitudinally at 30 μm and attached to poly-lysine-coated slides. After drying at room temperature, the sections were fixed in 4% PFA for 10 min and permeabilized in 0.5% TritonX-100 in PBS for 15 min at room temperature. The sections were incubated in α-BTX and DAPI in 1X PBS for an hour. Then, the EdU developing cocktail, comprising 1X PBS, 2% CuSO$_4$ (Sigma, 12849), sulfo-Cy3-azide (1 μg/ml; Lumiprobe, A1330), and ascorbic acid (20 mg/ml, Sigma, A4544) was added to the sections for 30 min, and then cover-slipped with Prolong Gold Antifade mounting medium. Confocal images were taken at 40X objective. On average, 15,000 myonuclei were sampled. Image analysis was performed using Imaris software as per method used for quantification of NMJ myonuclei.

**Acetylcholine receptor (AChR) turnover assay and analysis.** Acetylcholine receptor turnover was assessed by intramuscular injection of BTX-AF647 (25 pmol in 0.5 μl; Invitrogen, B35450) and BTX-AF488 (25pmol in 0.5 μl) into the tibialis anterior muscle on days 1 and 10, respectively. Mice were euthanized one day after the second injection. For the muscle damage group, 1.2% BaCl$_2$ solution (0.5 μl; Sigma, 342920) was injected into the tibialis anterior muscle to induce muscle damage 3 days prior to injection of first injection of BTX. The tibialis anterior muscle was dissected and sectioned at 40 μm in the longitudinal axis images. Images of AChRs were taken with the FV3000 Olympus confocal microscope at 60X objective with laser power and HV set to the highest setting without any pixel saturation. The images were then processed in Fiji software to calculate the fraction of pixels per image where the AF488 signal intensity was higher than that of AF647.

### Immunofluorescence staining, imaging, and analysis for MyHC

The soleus and tibialis anterior muscle were sectioned transversely at 15μm and attached to poly-lysine coated slides. After air-drying, the

sections were blocked with 10% goat serum in PBS for 1 h. Primary antibody cocktail comprising mouse IgG1-anti-MYH2 (1:60, Developmental Studies Hybridoma Bank), mouse IgG2b-anti-MYH7 (1:25, Developmental Studies Hybridoma Bank), mouse IgM-anti-and MYH4 (1:75, Developmental Studies Hybridoma Bank) was incubated 4ºC overnight, and then respective secondary antibodies (goat-anti-mouse IgG1-AF488 [1:400, Invitrogen, A-21121], goat-anti-mouse IgG2b-AF350 [1:200, Invitrogen, A-21140] and goat anti-mouse IgM-AF594 [1:400, Invitrogen, A-21044]) with Wheat Germ Agglutinin (WGA) Alexa Fluor™ 647 Conjugate (1 µg/ml, Invitrogen W32466) for 1 h. After staining was completed, the sections were coverslipped in Prolong Gold Antifade mounting medium. Slides were imaged at 20× magnification with stitching into whole sections using a TissueFAXS Slide Scanner (TissueGnostics) and TissueFAXSViewer software. The myofiber size of MyHC isoform populations was calculated using image J.

## Hematoxylin and eosin stain
The soleus and tibialis anterior muscle were sectioned transversely at 15 µm and attached to poly-lysine coated slides. After slides were defrosted, tissues were fixed in 4% PFA for 90 s, then rinsed in running water. Nuclei were stained with hematoxylin for 2 min, then destained with running water until the water was clear. Ammonia water was used for differentiation until nuclei were blue (10 s), then rinsed. Slides were stained with eosin (Sigma-Aldrich #HT110232) for 1 min. Dehydration was performed in a series of steps: 10 s in 75% ethanol, 30 s in 95% ethanol twice, 1 min in 100% ethanol twice. Slides were then subjected to Xylene for 30 s. Permount mounting solution was used to coverslip the slides, which were stored at RT. Slides were imaged at 20X with stitching into whole sections using a TissueFAXS Slide Scanner (TissueGnostics) and TissueFAXSViewer software.

## RNA analysis
RNA was extracted from muscle tissue using TRIzol (ThermoFisher Scientific, 10296028) according to the manufacturer's protocol. An amount of 1 µg of total RNA was reverse transcribed into cDNA using the High-Capacity cDNA Reverse Transcription Kits (Applied Biosystem, 4368813). Real-time PCR was performed in the Roche LightCycler® 480 Instrument II. Reactions were performed in triplicate, and relative amounts of cDNA were normalized to cyclophilin A (Ppia). The primers used in the present study are listed in Supplementary Table 2.

## RNAScope in situ hybridization
In situ hybridization was performed using the RNAScope V2 kit (Advanced Cell Diagnostics, 323100) to visualize the spatial distribution of mRNA of interest. The extensor digitorum longus muscle was sectioned at 20µm, and RNAScope was performed according to the manufacturer's protocol for fresh frozen sections. In brief, fresh frozen EDL was sectioned longitudinally at 20 µm, fixed in 4% PFA, dehydrated in increasing concentration of ethanol, and then incubated with $H_2O_2$ provided in the kit to sequester endogenous peroxidase activity. After which, the sections were treated with kit-provided Protease IV, followed by incubation with the respective RNAScope probes (Supplementary Table S3) at 40 °C for 2 h. The sections were stored in 5X SSC buffer (1st Base, BUF-3051) overnight, and the amplification and multiplexing steps took place the next day. Multiplexing was performed using Opal multiplex detection kits (Akoya Bioscience; Opal 520 [FP1487001KT], Opal 570 [FP1488001KT] and Opal 690 [FP1497001KT]). Upon completion of the amplification and multiplexing steps, the sections were incubated with DAPI and cover-slipped with Prolong Gold Antifade. The sections were imaged within one week of staining using the FV3000 confocal microscope at 100X objective.

## Protein analysis and Western blotting
Gastrocnemius muscle from 4-month-old mice was homogenized in cold SDS lysis buffer (50 mM Tris, pH 7.5, 70 mM urea, 250 mM sucrose, and 2% SDS) with Protease Inhibitor Cocktail (Roche, 4693159001) and Phosphatase Inhibitor Cocktail II (Sigma, P5726) and III (Sigma, P0044). 30 µg of total protein was separated in 10% polyacrylamide gel (Bio-Rad, #1610158) and transferred to nitrocellulose membranes. Ponceau red (MP Biomedicals, 02190644) was used to check the transfer efficiency and also applied to the total protein loading. After washing out the ponceau red staining, the membrane was blocked with 5% milk for an hour. The following antibodies from Cell Signaling were used for Western blotting: α- p70 S6 kinase 1 (9202), α- p70 S6 kinase 2 (14130), α-S6 Ribosomal Protein (2217), α-phospho-S6 ribosomal protein Ser240 + 244 (2215), α– 4EBP1 (9452), α phospho-4EBP1 Thr37 + 46 (2855), α- Akt (4691), α phospho-AKT Ser473(4060) and α-actinin (3134). α phospho-p70 S6 kinase 2 Ser423 (SAB4301595) was purchased from sigma and α TSC1 (A300-316A) was purchased from Axil Scientific.

## Endplate potential recordings
The diaphragm with the phrenic nerve attached was quickly dissected from the mouse and bathed in oxygenated Ringer's solution containing (in mM): 10 Glucose, 125 NaCl, 25 NaHCO₃, 1.25 NaH₂PO₄.2H₂O, 2.5 KCl, 1.8 CaCl₂, 1 MgCl₂, pH 7.4 (300−310 mOsm). Under the microscope, the nerve was drawn into the suction electrode by applying gentle suction. Muscle contraction to nerve stimulation was tested by applying a single pulse of stimulation to ensure the viability of the phrenic nerve-diaphragm preparation. Stimulation was performed with the A365 stimulation isolator (WPI). Muscle contraction was subsequently blocked using the muscle sodium channel blocker, µ-conotoxin GIIIb (2 µM; Alomone, C-270). Spontaneous miniature endplate potential (mEPP) and nerve-evoked endplate potential (EPP) recordings were made at room temperature under the current clamp with multi-clamp 200B amplifier. The recording electrode solution is made up of 2 M potassium gluconate and 10 mM KCl. EPP and mEPP recordings were started 30 min after the diaphragm was placed in the bath solution. Only fibers with a stable resting membrane potential ≤ −60 mV were analyzed. On average, 8 myofibers were recorded per diaphragm.

## Behavioral testing
Motor coordination function was assessed on the rotarod. The mice were placed on the accelerating rotarod (Ugo Basile) starting at 4 rounds per min (RPM) and accelerated to 40 RPM over 5 min. The time spent on the rotarod was recorded, and the trial would stop when the mouse fell off or spent a maximum of 10 min on the rotarod. Each animal was given 4 trials per day and the test was performed over 3 consecutive days. Neuromuscular strength was evaluated by four limbs hanging tests, which requires balance and grip strength[36,37]. Mice were placed on a grid, which was then inverted above a cage filled with bedding. The session ended after a hanging time for 600 s was achieved. The maximum hanging time from three sessions was used for further analysis.

## Statistical analysis
Unless otherwise stated, all results are expressed as mean ± SEM of independent animals. A two-tailed unpaired Student's t-test was used to determine the statistical significance between the two groups. The significance of differences between multiple groups was evaluated by either one-way or two-way ANOVA followed by Tukey post-hoc pairwise comparison. Statistical analyses were performed using GraphPad Prism 9 software. A P value of less than 0.05 was considered statistically significant.

## Study approval
All experimental procedures involving the use of animals were reviewed and approved by the IACUC of the National University of Singapore.

**Reporting summary**

Further information on research design is available in the Nature Portfolio Reporting Summary linked to this article.

## Data availability

All data presented in this study are available from the corresponding authors upon reasonable request. The source data underlying figures and Supplementary figures are provided in the separated Source Data file. Source data are provided with this paper.

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

## Acknowledgements

This study was supported by grants from National Medical Research Council OFIRG18nov-0093 (S.Y.T.), Ministry of Education Tier1 NUHSRO/2018/012/T1 (S.Y.T.), SoM Start-Up Grant NUHSRO/2016/097/SU/01 (S.Y.T.), Medical Sciences Cluster Autophagy Seed Grant (S.Y.T.), and NUHS-CMU Collaboration Grant NUHSRO/2019/CMU/002 m (S.Y.T.). Work was supported by Medical Research Council grant MC-A654-5QB40 and a Wellcome Trust Grant 098565/Z/12/Z in D.J. Withers' lab. We would like to express our gratitude to D.J. Ham, M. Kaeberlein, B.K. Kennedy, T.W. Koh, S. Lin and M.A. Rùegg for their invaluable feedback on the manuscript; also special thanks to S.Y. Lee and D. Pang of the Confocal Microscopy Unit, NUS, for their advice on image acquisition and analysis.

## Author contributions

S.T.J.A., E.M.C., K.T.T., A.H., D.Y., H.D., S.K., and H.H. performed the experiments. S.T.J.A., H.D., and H.H. analysed the data. S.A., E.M.C., and D.J.W. contributed the mouse models and muscle samples. S.T.J.A. and S.Y.T. developed the concepts, supervised the work, and wrote the manuscript. S.Y.T. secured the funding. All authors contributed to the editing of the manuscript.

## Competing interests

The authors declare no competing interests.
