## [Peer Review File · Nature Communications]

Muscle 4EBP1 activation modifies the structure and function of the neuromuscular junction in miceREVIEWER COMMENTS

Reviewer #1 (Remarks to the Author):

In this manuscript by Ang et al., the authors examine the role of mTORC1 signaling on the NMJ. While they examined some interesting aspects, like the number of nuclei at the NMJ and newly generated mouse models, the analyses leave a lot of questions.

Major issues:

- The use of TSC1 ko mice at 12 month of age makes it impossible to link the phenotype back to mTORC1 signaling. These mice have already a very severe phenotype at this point (Castets 2013), making it very difficult to determine cause and effect. Mice at 12 months of age have only 25% of muscle force left and lost a lot of weight, making it impossible to understand what is behind the observed alterations.
- The authors do not show any histological characterization (e.g. H&E staining) of the different mouse models used, especially of aged S6K1mko and 4E-BP1mt-muscle or TSC1ko-S6K1mko and TSC1ko-4EBP1mt. In addition, also a western blot for Akt-mTORC1 signaling should be done on these mice.
- They performed electromyography analysis only on 3-month-old 4EBP1-mt diaphragms, where they found enhanced neuromuscular transmission despite NMJ fragmentation. However, they analyzed the number of AchR fragments at 6 months of age in these mice.
- The authors speculate on the fact that during ageing the fibers positive for P-S6 are denervated, because of their angulated shape. However, staining for P-S6 and NCAM should be done on serial sections in order to understand if fibers positive for P-S6 are also positive for NCAM. This should be done also in TSC1ko, TSC1ko-S6K1mko and TSC1ko-4EBP1mt where the shape of P-S6-positive myofibers do not remind to the one of denervated fibers.
- It is known that NCAM is used as a marker for fibers denervation, but the time course of its expression in young and old muscles upon denervation is a bit controversial. Thus, Mander's coefficient analysis to measure muscle innervation should be done in all the mouse models. In addition, mRNA expression of other markers for muscle denervation, such as Musk, Noggin, Runx1, Myogenin etc. should be done.
- The number of mice/groups in RT-PCR experiments should be increased.
- The authors' main result is the enhanced neurotransmission observed in 4EBP1mt-muscle mice. What about muscle function in these animals? And also, muscle force of TSC1ko mice is partially rescued by S6K1 deletion or expression of mutant 4EBP1?
- S6K1 deletion slightly attenuates age-induced NMJ fragmentation, but has no effect when mTORC1 is over-activated. So, they conclude that S6K1 has no effect on NMJ instability during ageing or in TSC1ko mice, whereas 4EBP1 is more involved in NMJ homeostasis. However, S6 is still highly phosphorylated in these animals, due to compensatory effects of S6K2. Thus, a double deletion for S6K1 and 2 should be done to better elucidate the role of S6Ks-S6 axis in NMJ stability.
- The authors focused on number of myonuclei at NMJ level in the different mouse models. They found that it is increased in 4EBP1mt-muscle, together with increased satellite cells proliferation, resulting in augmented AchR turnover and NMJ fragmentation. However, the number of myonuclei/NMJ is changing in all the mouse models examined. Indeed, aged males show decreased myonuclei as females and 12-month-old mice. But, NMJ fragmentation is observed only in aged males. In addition, also aged S6K1mko mice show less myonuclei and less AchR fragments. Have you an idea of cell proliferation in the other mouse models? Is there a direct correlation between number of myonuclei/NMJ and cell proliferation/AchR turnover?

Minor comments:

- In Figure legend 1E: they explained what solid arrows, open arrows and asterisks stand for. However, in the figure 1E, only open arrows, asterisks and hash are found.

- line 393: 4EBP1mt-mice mice should be corrected.

- line 425: froze should be changed in frozen.

Reviewer #3 (Remarks to the Author):

The manuscript "Muscle 4EBP1 activation..." reports results of a study designed to elucidate how alterations of the NMJ explain the sarcopenia that affects the aged. It has been shown previously that disruption of mTOR affects structure and function of the NMJ and contributes to age-related loss of muscle mass and strength. What has yet to be clearly established, however, are the downstream mechanisms by which reduced mTOR production disrupts NMJ structure and function. Here, the authors examine the interaction of two of these downstream targets – S6K1 and 4EBP1-with regulation of the NMJ's morphology and function. Ultimately, the authors wish to find a way to prevent age-related NJ decay, thus preventing sarcopenia and its many associated health concerns. This is an interesting and novel approach to countering sarcopenia and the results presented here are, in fact, of real interest. That said, there are concerns with some of the comments and assumptions made here that prevent it from being acceptable for publication in Nature Communications, in its current form.

1) The presence of "fragmented" post-synaptic endplates should not be assumed to be indicative of damage or dysfunction. Several studies have shown that such endplate fragmentation, or dispersion, occurs as a result of increased neuromuscular activity such as that associated with chronic endurance exercise training (e.g., Deschenes et al, 1993).

2) There is some concern about using, by itself, increased NCAM expression as an accurate marker of denervation. Is there any other evidence of denervation here? It is mentioned that gamma subunit expression, but only in passing, perhaps this should be examined more carefully.

3) It seems of great interest that there were such significant sex-related differences in evidence of structural disturbance of NMJs in that male, but not female mice showed endplate fragmentation with aging. But rather than further pursuing this seemingly rich topic for greater insight, authors elect to simply drop female mice from further study, and choose to focus only on males for further investigation. Please explain why.

4) Does NCAM expression relates with myonuclei expression at endplates? What might this indicate?

5) In lines 335-336, it is described that 4EBP1 deletion resulted in extreme endplate structural remodeling, but that neuromuscular transmission was unaffected. What does this say about the biological tenet that form and function are inextricably linked? Please address.

6) An important finding of this work is how changes in downstream targets for mTOR affect myofiber type expression, and in turn, neuromuscular transmission. In view of this, it would have been insightful to stain for specific fiber type expression (Type I, IIA, IIX, and IIB) muscle examined.

Immunofluorescent staining of the soleus, EDL and TA muscles is recommended.

We thank the editor and the reviewers for their thoughtful comments and have modified the manuscript accordingly. The manuscript has improved as a result, and we hope you find the changes satisfactory. In the present work, we showed that activating mTORC1-4EBP1 signaling induces NMJ renewal by expanding the pool of post-synaptic myonuclei and increasing AChR turnover as a novel intervention to mitigate sarcopenia. We appreciate that you recognize the novelty of our approach in countering sarcopenia.

Following your advice, we have incorporated the following new data into our manuscript:

1. Representative images of serial sections stained with pS6^{S240+244} or NCAM from cross-sections of the gastrocnemius muscle are now included in **Fig.2c** to confirm their colocalization in 28-month-old male control and 12-month-old male TSC1mKO mice.
2. Quantifications of NMJ have now added female counterparts for all the genetic groups, including in **Fig.3c-e**, **4d-f**, **S1d**, and **S4b-d**.
3. Mander's colocalization coefficient measurements by the degree of overlap between pre- (neurofilament and synaptophysin) and post-synaptic (BTX) components have been added to all genetic groups, including in **Fig.3e**, **4f**, and **S4d**.
4. RT-PCR analysis for youth (6-month-old) versus nature aging (28-month-old) mouse cohort and 12-month-old mouse cohorts have now added female counterparts, including in **Fig.S1b**, **S1e**, and **S5f-g**. The number of mouse samples and genes for analysis is also increased. Additionally, S6K1-TSC1mKO mouse muscle samples have now been included in **S5f** and **g**.
5. Western blot analysis of mTORC1 signaling from control, TSC1mKO, S6K1-TSC1mKO, and 4EBP1mt-TSC1mKO mice are present in **Fig. S1c** and **S5h**.
6. Myofiber analysis by H&E and MyHC from young (6-month-old) and old (28-month-old) S6K1mKO mice are present in **Fig. S2**.
7. Myofiber analysis by H&E from 12-month-old control, TSC1mKO, S6K1-TSC1mKO, and 4EBP1mt-TSC1mKO mouse are present in **Fig. S6**.
8. The staining of NMJ from 3-month-old control and 4EBP1mt-muscle muscle are present in **Fig. S7a**.

Change of Figure Formatting:

1. Fig.1f and 3f: Quantification of the mean CSA of NCAM- positive (NCAM+) versus NCAM-negative (NCAM-) fibers in the gastrocnemius muscle.

In order to show and emphasize the comparison of CSA between NCAM- positive (NCAM+) versus NCAM-negative (NCAM-) fibers, we re-order the bars.

2. Fig. 2b and S3c: Quantification of mean CSA of pS6^{S240+244} -positive (pS6+) versus pS6^{S240+244} -negative (pS6-) fibers (right panel) in the gastrocnemius muscle.

In order to show and emphasize the comparison of CSA between pS6- positive (pS6+) versus pS6-negative (pS6-) fibers, we re-order the bars.

Reordering on figures:

1. Fig. 2e has become Fig. 2f.
2. Fig. 4f has become Fig. 4g.
3. Fig. S1c and S1d have become Fig. S1d and S1e, respectively.
4. Fig. S2 has become Fig. S3.
5. Fig. S3a, S3b, S3c, S3d, and S3e have become Fig. S4a, S4b, S4c, S4e, and S4f, respectively.
6. Fig. S4a, S4b, S4c, S4d, S4e, and S4f have become Fig. S5a, S5b, S5c, S5d, S5e, and S5g, respectively.
7. Fig. S5a, S5b, S5c, and S5d have become Fig. S7b, S7c, S7d, and S7e, respectively.

Responses to individual reviewers:

REVIEWER COMMENTS

Reviewer #1 (Remarks to the Author):

In this manuscript by Ang et al., the authors examine the role of mTORC1 signaling on the NMJ. While they examined some interesting aspects, like the number of nuclei at the NMJ and newly generated mouse models, the analyses leave a lot of questions.

Major issues:

- The use of TSC1 ko mice at 12 month of age makes it impossible to link the phenotype back to mTORC1 signaling. These mice have already a very severe phenotype at this point (Castets 2013), making it very difficult to determine cause and effect. Mice at 12 months of age have only 25% of muscle force left and lost a lot of weight, making it impossible to understand what is behind the observed alterations.

As reviewer #1 points out, TSC1mKO mice have accelerated sarcopenia development, and 21 days of rapamycin treatment were able to attenuate the myopathy and restore muscle function (Castets 2013 doi: 10.1016/j.cmet.2013.03.015). The same group later published that NMJs of adult TSC1mKO (12-month-old) have shown similar severity of NMJ instability to natural aging NMJs (30-month-old). Similarly, one-month and long-term rapamycin treatment (~10-15 months) attenuated NMJ instability in adult TSC1mKO and aged control mice, respectively (Ham 2020 doi: 10.1038/s41467-020-18140-1), suggesting mTORC1 hyperactivation is the underlying mechanism of NMJ instability during aging.

Following their finding, we further analyze which downstream effector of mTORC1, known to be targeted by rapamycin, could contribute to NMJ instability in adult TSC1mKO mice (12-month-old) and aged mice (28-month-old). We expected that the genetic manipulation of mTORC1 downstream effectors would have more substantial effects on the regulation of NMJ stability. This was the rationale for the study.

To our surprise, muscular inactivation of S6K1 had a very mild phenotype in the NMJ maintenance, while muscular hyper-activation of 4EBP1 drove dramatic NMJ remodeling.

- The authors do not show any histological characterization (e.g. H&E staining) of the different mouse models used, especially of aged S6K1mko and 4E-BP1mt-muscle or TSC1ko-S6K1mko and TSC1ko-4EBP1mt. In addition, also a western blot for Akt-mTORC1 signaling should be done on these mice.

The histological analysis of 4EBP1mt-muscle mouse muscle from young and old mice was published in the previous report (Tsai 2015 doi: 10.1172/JCI77361). The histological analysis of young versus old S6K1mKO mouse muscle is now presented in Fig. S2. The histological analysis of adult TSC1mKO, S6K1-TSC1mKO, and 4EBP1mt-TSC1mKO mouse muscles are included in Fig. S6.

The western blot of Akt-mTORC1 signaling from adult TSC1mKO, S6K1-TSC1mKO, and 4EBP1mt-TSC1mKO mouse muscle is now presented in Fig. S5h.

- They performed electromyography analysis only on 3-month-old 4EBP1-mt diaphragms, where they found enhanced neuromuscular transmission despite NMJ fragmentation. However, they analyzed the number of AchR fragments at 6 months of age in these mice.

We have included the image of NMJ staining from 3-month-old 4EBP1mt-muscle mice (Fig. S7a), in which grape-like aggregates of AChR have appeared, confirming the fragmentation of AChR.

- The authors speculate on the fact that during ageing the fibers positive for P-S6 are denervated, because of their angulated shape. However, staining for P-S6 and NCAM should be done on serial sections in order to understand if fibers positive for P-S6 are also positive for NCAM. This should be done also in TSC1ko, TSC1ko-S6K1mko and TSC1ko-4EBP1mt where the shape of P-S6-positive myofibers do not remind to the one of denervated fibers.

Representative images of serial sections stained with pS6^{S240+244} or NCAM from cross-sections of the gastrocnemius muscle are now included in Fig.2c to confirm their colocalization in 28-month-old male control and 12-month-old male TSC1mKO mice.

- It is known that NCAM is used as a marker for fibers denervation, but the time course of its expression in young and old muscles upon denervation is a bit controversial. Thus, Mander's coefficient analysis to measure muscle innervation should be done in all the mouse models. In addition, mRNA expression of other markers for muscle denervation, such as Musk, Noggin, Runx1, Myogenin etc. should be done.

Mander's colocalization coefficient measurements have now been added to all genetic groups, including in Fig.3e, 4f, and S4d.

Real-time PCR analysis on genes known to be expressed in NMJ (Musk and Lrp4), to induce muscle atrophy upon denervation (Runx1 and Gadd45a), and to be involved in muscle regeneration/differentiation (Noggin and Myogenin) upon denervation are included in Fig. S1b, S1e, and S5f,g.

- The number of mice/groups in RT-PCR experiments should be increased.

We have included female counterparts and increased the number of mouse samples for RT-PCR analysis in Fig.S1b, S1e, and S5f, g. Additionally, S6K1-TSC1mKO mouse muscle samples are included in Fig. S5f,g.

- The authors' main result is the enhanced neurotransmission observed in 4EBP1mt-muscle mice. What about muscle function in these animals? And also, muscle force of TSC1ko mice is partially rescued by S6K1 deletion or expression of mutant 4EBP1?

For the 4EBP1mt-muscle mouse, we did not observe any significant difference in muscle performance between transgenic mice and their littermate controls, such as treadmill (Tsai 2015 doi: 10.1172/JCI77361) or rotarod (this study Fig. 5a). We did not have a device in Singapore that could measure the nerve-stimulated muscle force for the physiological readout of NMJ. Since this assay requires freshly harvested muscle, it was not feasible to find an overseas collaborator for this experiment in three months' timeframe.

We collaborated with Prof. Julien Ochala from the University of Copenhagen to perform an indirect measurement of the muscle force by the actin-myosin interaction *in vitro* without involving excitation-contraction coupling. The preliminary data shows no significant difference in the actin-myosin interaction between age-matched littermate control versus 4EBP1mt-muscle mouse muscle (Fig. R1A).

The full analysis of muscle function from S6K1-TSC1mKO and 4EBP1mt-TSC1mKO is prepared in another manuscript we are working on now. The preliminary data from the measurement of the actin-myosin interaction shows that activation of 4EBP1 could restore the decline of actin-myosin interaction in TSC1mKO mouse background, while inactivation of S6K1 has no protection effect (Fig. R1B). Since the current manuscript focuses on the NMJ instability induced by mTORC1 hyper-activation in natural aging and genetic mouse model, we did not include the analysis of muscle function to diffuse this manuscript's key message and fear that it could complicate the main conclusions of the story.

- S6K1 deletion slightly attenuates age-induced NMJ fragmentation, but has no effect when mTORC1 is over-activated. So, they conclude that S6K1 has no effect on NMJ instability during ageing or in TSC1ko mice, whereas 4EBP1 is more involved in NMJ homeostasis. However, S6 is still highly phosphorylated in these animals, due to compensatory effects of S6K2. Thus, a double deletion for S6K1 and 2 should be done to better elucidate the role of S6Ks-S6 axis in NMJ stability.

We agree that the potential compensatory effects of S6K2 are important to fully access the S6Ks-S6 axis in NMJ stability. Yet, we could not have done this experiment in a short time since S6K2 flox mice, or S6K2 whole-body knockout mice, are not available to us.

S6K1 has previously been proposed as a dominant S6K isoform to regulate growth in skeletal muscle independent of S6K2 (Ohanna 2005, doi: 10.1038/ncb1231). Whole-body knockout of S6K1 female mice had better muscle performance with age and were long-lived (Selman 2009, doi: 10.1126/science.1177221). Additionally, muscle-specific deletion of S6K1 is sufficient to extend lifespan in *LMNA* knockout mice, a genetic model for the study of laminopathy, to a similar degree as rapamycin treatment (Liao 2017, doi: 10.1038/celldisc.2017.39.). Yet, surprisingly, we found that muscle-specific deletion of S6K1 has no effect on skeletal muscle growth or maintenance with age. S6K1mKO mice have comparable muscle type distribution and size to control mice (Fig. S2). Muscle-specific deletion of S6K1 also does not rescue myopathy, such as inclusions and degenerated basophilic fibers in the TSC1mKO mouse background (Fig. S6).

- The authors focused on number of myonuclei at NMJ level in the different mouse models. They found that it is increased in 4EBP1mt-muscle, together with increased satellite cells proliferation, resulting in augmented AchR turnover and NMJ fragmentation. However, the number of myonuclei/NMJ is changing in all the mouse models examined. Indeed, aged males show decreased myonuclei as females and 12-month-old mice. But, NMJ fragmentation is observed only in aged males. In addition, also aged S6K1mko mice show less myonuclei and less AchR fragments. Have you an idea of cell proliferation in the other mouse models? Is there a direct correlation between number of myonuclei/NMJ and cell proliferation/AchR turnover?

The correlation between post-synaptic myonuclei numbers and satellite cell activity in the regulation of NMJ stability was reported by Chakkalakal's group (Liu 2017, doi: 10.7554/eLife.26464). Similar to our finding, their control mice at 12 months and 24 months also had a comparable number of myonuclei. Still, a significant post-synaptic degeneration (quantified by branch morphology of AChR-enriched area) progressed further at 18 and 24 months. Depleting satellite cells reduces post-synaptic myonuclei number and accelerates post-synaptic degeneration in 12-month-old mouse muscles. Moreover, satellite cell depletion causes more profound reductions in NMJ reinnervation and post-synaptic myonuclei number upon sciatic nerve transection. They did not perform the AChR turnover experiment, so we were unsure whether there was a relationship between satellite cell activities and AChR turnover. Nonetheless, their data suggests that satellite cell activities (proliferation/incorporation to myonuclei at NMJ) are crucial for neuromuscular regeneration and maintenance. The reduction of post-synaptic myonuclei number precedes the fragmentation of AChR clusters with age and decreased post-synaptic myonuclei number accelerates the fragmentation of AChR clusters in adult mice.

There was evidence suggesting sex dimorphism regulation in satellite cell activities. For example, female satellite cells have greater self-renewal and regenerative capacity than their male counterparts (Deasy 2007, doi: 10.1083/jcb.200612094). Even though the number of post-synaptic myonuclei is reduced at the same rate, the higher regenerative capacity of female satellite cells could contribute to the slower progression of AChR fragmentation in the older females. More studies are needed to verify the sex dimorphism in muscle aging.

Regarding myofiber mTORC1 signaling regulating satellite cell activity and NMJ stability, a previous study on TSC1mKO mice upon nerve injury (Castets 2019, doi:10.1038/s41467-019-11227-4) showed that the expressions of myogenin and MyoD, the activated myogenic differentiation markers of myoblast/satellite cells, were delayed and reduced, suggesting that satellite cell activation was decreased. AChR turnover was also impaired in these denervated TSC1mKO mouse muscles.

Our study found that endogenous 4EBP1 is highly expressed in the post-synaptic myonuclei. Activation of 4EBP1 induces NMJ regeneration by increased post-synaptic myonuclei number recruitment from satellite cell activation and promoted AChR turnover without stress. Together with the finding in TSC1mKO mice, we proposed that myofiber mTORC1-4EBP1 signaling is critical for NMJ regeneration by regulating satellite cell activities and AChR turnover.

We didn't analyze satellite cell activities, and AChR turnover in S6K1mKO mice since their phenotype of NMJ stability is very subtle. Currently, we don't have aged S6K1mKO mice in house. Therefore, it is not feasible to perform these experiments in three months' timeframe.

Minor comments:

- In Figure legend 1E: they explained what solid arrows, open arrows and asterisks stand for. However, in the figure 1E, only open arrows, asterisks and hash are found.

Thanks for pointing out our mistake. We have corrected the Figure legend 1e.

- line 393: 4EBP1mt-mice mice should be corrected.

Thanks for pointing out our mistake. We have corrected the text.

- line 425: froze should be changed in frozen.

Thanks for pointing out our mistake. We have corrected the text.

Reviewer #3 (Remarks to the Author):

The manuscript "Muscle 4EBP1 activation..." reports results of a study designed to elucidate how alterations of the

NMJ explain the sarcopenia that affects the aged. It has been shown previously that disruption of mTOR affects structure and function of the NMJ and contributes to age-related loss of muscle mass and strength. What has yet to be clearly established, however, are the downstream mechanisms by which reduced mTOR production disrupts NMJ structure and function. Here, the authors examine the interaction of two of these downstream targets – S6K1 and 4EBP1-with regulation of the NMJ's morphology and function. Ultimately, the authors wish to find a way to prevent age-related NJ decay, thus preventing sarcopenia and its many associated health concerns. This is an interesting and novel approach to countering sarcopenia and the results presented here are, in fact, of real interest. That said, there are concerns with some of the comments and assumptions made here that prevent it from being acceptable for publication in Nature Communications, in its current form.

1) The presence of “fragmented” post-synaptic endplates should not be assumed to be indicative of damage or dysfunction. Several studies have shown that such endplate fragmentation, or dispersion, occurs as a result of increased neuromuscular activity such as that associated with chronic endurance exercise training (e.g., Deschenes et al, 1993).

Deschenes's group showed that increased the area of the endplate upon endurance exercise training (Deschenes 2011, doi: 10.1016/j.neuroscience.2011.05.070.; Deschenes 2016, doi: 10.1016/j.neuroscience.2015.12.004.) and their 2-D images of BTX staining reflected AChR cluster distribution also seem to be fragmented (Deschenes 1993, doi: 10.1007/BF01181487.; Deschenes 2019, doi: 10.1016/j.cophys.2019.02.004). Moreover, Spitsbergen's group observed an increased endplate area in the rat following 2 weeks of training independently (Gyorkos, 2014, doi: 10.1016/j.neuroscience.2013.10.068.). Yet, neither group mentioned the discontinued feature of AChR cluster nor quantified the fragmentation in their exercise samples. They have also not followed up on whether the increased AChR cluster distribution is due to an increased AChR turnover. We are hesitant to cite their findings based on our interpretation.

Our data suggested that the dramatic fragmentation of endplate in 4EBP1mt-muscle mouse is contributed from activated muscle and NMJ regeneration, as we stated in the last section of result: “Overall, these data indicate that muscle regeneration and NMJ turnover were enhanced, which both attribute to inducing drastic morphological changes and enhancing neurotransmission at the NMJ of 4EBP1mt-muscle mouse.” We have re-emphasized our hypothesis in the discussion: “The fragmentation phenotype driven by chronic 4EBP1 activation is likely a result of cycles of degeneration and regeneration of the myofibers and NMJs.” Similar phenomena have been observed in the muscle regeneration model that uses laser ablation to damage myofiber specifically without interfering with motor neuron connection. Thompson's group showed newly synthesized AChR appears after muscle regeneration (Li 2011, doi:10.1523/JNEUROSCI.2953-11.2011).

Endurance exercise attenuated aging-associated decline in satellite cell content (Shefer 2010, doi: 10.1371/journal.pone.0013307.t001; Shefer 2013, doi: 10.1111/febs.12228.; and Cisterna 2016, doi: 10.1111/joa.12429) and activities upon muscle degeneration (Joanisse 2016, doi: 10.1096/fj.201600143RR; Brett 2020, doi: 10.1038/s42255-020-0190-0). Whether endurance exercise-induced satellite cell activation could directly contribute to AChR renewal remains unknown and would be of great interest for future studies.

2) There is some concern about using, by itself, increased NCAM expression as an accurate marker of denervation. Is there any other evidence of denervation here? It is mentioned that gamma subunit expression, but only in passing, perhaps this should be examined more carefully.

Ruegg's group has published a detailed structural analysis of denervation in adult TSC1mKO (Ham 2020 doi: 10.1038/s41467-020-18140-1). Our NCAM IF further supported the structural abnormality in these mice identified previously. While NCAM expression might be up-regulated not only by degeneration/denervation but also by regeneration or activated NMJ remodeling. The staining pattern of NCAM would be more suitable to differentiate degenerated/denervated fibers (the intracellular accumulation of NCAM) versus fibers with activated NMJ remodeling (junctional NCAM).

Real-time PCR analysis on atrophy-related genes (Runx1 and Gadd45a) induced upon denervation are now included in Fig. S1b, S1e, and S5f,g. Along with the up-regulation of Chrng and down-regulation of Chrne (adult/mature form of AChR), Runx1 and Gadd45a are also increased in adult TSC1mKO mice.

3) It seems of great interest that there were such significant sex-related differences in evidence of structural disturbance of NMJs in that male, but not female mice showed endplate fragmentation with aging. But rather than further pursuing this seemingly rich topic for greater insight, authors elect to simply drop female mice from further study, and choose to focus only on males for further investigation. Please explain why.

Quantifications of NMJ have now added female counterparts for all the genetic groups, including in Fig.3c-e, 4d-f, S1d, and S4b-d. RT-PCR analysis for youth (6-month-old) versus nature aging (28-month-old) mouse cohort, and 12-month-old mouse cohorts have now added female counterparts, including in Fig.S1b, S1e, and S5f-g. Noted differences in male and female NMJs stability that respond differently to aging were not observed in the TSC1mKO or 4EBP1mt-muscle mouse model, suggesting mTORC1 hyperactivation or 4EBP1 activation could drive NMJs remodeling.

4) Does NCAM expression relates with myonuclei expression at endplates? What might this indicate?

The intracellular NCAM staining from aged control and indicated transgenic mouse lines supposedly are from its expression in myonuclei. Ruegg's group has published a gene expression profile using laser-capture microdissection to differentiate gene expression from myonuclei in NMJ region (surround aBTX stained positive area) versus non-NMJ regions in male adult (10 months) versus sarcopenic (30 months) muscle (Ham 2020 doi: 10.1038/s41467-020-18140-1). The expression of NCAM is enriched in the NMJ region versus the non-NMJ region. Due to the heterogeneous aging process, there is no significant increased NCAM expression in either region compared to adult versus sarcopenic mouse muscles. Whether more post-synaptic myonuclei will increase total NCAM expression in myofiber? To test the correlation, it will require single-cell sequencing and a way to manipulate the number of post-synaptic myonuclei and non-synaptic myonuclei artificially.

Nonetheless, using published SarcoAtlas from Ruegg's group (<https://sarcoatlas.scicore.unibas.ch/>), the expression of NCAM is also upregulated in their male TSC1mKO mouse (9 months old), and rapamycin treatment seems to repress NCAM expression slightly (Fig. R2). Similar to our finding, the expression of NCAM is upregulated upon the myofiber mTORC1 signaling-caused denervation regardless of post-synaptic myonuclei number.

5) In lines 335-336, it is described that 4EBP1 deletion resulted in extreme endplate structural remodeling, but that neuromuscular transmission was unaffected. What does this say about the biological tenet that form and function are inextricably linked? Please address.

We would like to clarify that 4EBP1mt-muscle mice are not 4EBP1 knockout mice. The 4EBP1mt-muscle mouse is a transgenic mouse line that over-expression of a mutant form of 4EBP1 within the mTOR phosphorylation site (threonine to alanine at amino acid positions 37 and 46) to yield a constitutively active form of the protein that is non-responsive to mTORC1 regulation in differentiated myofiber.

The enhancement of neuromuscular transmission was observed in 4EBP1mt-muscle mice, which 4EBP1 hyperactivation is restricted in myofiber, indicating an increase in post-synaptic sensitivity to quantal release. We hypothesize that since the AChRs of 4EBP1mt-muscle NMJs were highly fragmented and dispersed over a larger endplate area, which might have resulted in a higher magnitude of post-synaptic membrane depolarization. Hence, we found upregulation of AChR genes in 4EBP1mt-muscle mice and expansion of endplate area, which could imply a higher number of AChR receptors. Another possible explanation for enhanced neurotransmission in 4EBP1mt-muscle mice is due to changes in fiber type composition. The electrophysiological profiles of fast-twitch glycolytic and slow-twitch oxidative fibers are different. For example, fast-twitch EDL and tibialis anterior have lower mEPP or mEPC than slow-twitch soleus. We have shown that increased slow-twitch oxidative fibers presented in 4EBP1mt-muscle mice (Tsai 2015 doi: 10.1172/JCI77361). Thus, fiber type transformation to an oxidative phenotype might explain the enhanced neuromuscular transmission in 4EBP1mt-muscle mice. In the same vein, 4EBP1mt-muscle NMJs demonstrated improved transmission fatigability to repeated nerve stimulations that are likely due to such fiber type transformation.

6) An important finding of this work is how changes in downstream targets for mTOR affect myofiber type expression, and in turn, neuromuscular transmission. In view of this, it would have been insightful to stain for specific fiber type expression (Type I, IIA, IIX, and IIB) muscle examined. Immunofluorescent staining of the soleus, EDL and TA muscles is recommended.

The histological analysis of 4EBP1mt-muscle mouse muscle from young and old mice was published in the previous report (Tsai 2015 doi: 10.1172/JCI77361). The histological analysis of young versus old S6K1mKO mouse muscle is now presented in Fig. S2. The H&E of adult TSC1mKO, S6K1-TSC1mKO, and 4EBP1mt-TSC1mKO mouse muscles are included in Fig. S6. The full analysis of MyHC from S6K1-TSC1mKO and 4EBP1mt-TSC1mKO is being investigated in our ongoing study. The preliminary data shows that the distribution of each fiber type is not changed in TSC1mKO mice, consistent with the previous report (Castets 2013 doi: 10.1016/j.cmet.2013.03.015). Activation of 4EBP1 or inactivation of S6K1 in the TSC1mKO mouse background also does not change the portion of each fiber type expression, yet 4EBP1 activation is able to restore type IIb atrophy in the TSC1mKO mouse background.

REVIEWER COMMENTS

Reviewer #1 (Remarks to the Author):

The authors have significantly improved the previous manuscript. I personally believe muscle function is a critical readout to be added to the manuscript, as most models of muscle dysfunction are linked to NMJ alterations. So having an idea on muscle function in the different transgenic animals will allow for a better understanding of the factors which might affect directly NMJ morphology/function.

Reviewer #2 (Remarks to the Author):

In general, the authors have done a fine job in addressing concerns expressed with initial iteration of this paper. Some concerns remain about misunderstanding about fragmentation necessarily indicating damage to NMJ and rather just as an indicator of normal NMJ remodeling. But besides that, fine job in answering my earlier concerns.

We thank reviewers for their comments. We have revised our manuscript according to the comments as indicated below.

Responses to individual reviewers:

REVIEWER COMMENTS

Reviewer #1 (Remarks to the Author):

The authors have significantly improved the previous manuscript. I personally believe muscle function is a critical readout to be added to the manuscript, as most models of muscle dysfunction are linked to NMJ alterations. So having an idea on muscle function in the different transgenic animals will allow for a better understanding of the factors which might affect directly NMJ morphology/function.

We have now included four limbs hanging tests as an assessment for muscle function in Fig. S2g (S6K1mKO mice) and Fig. S6c (TSC1mKO, S6K1-TSC1mKO, and 4EBP1mt-TSC1mKO mice). In concordance with muscle size, S6K1mKO mice have similar hanging times to their control littermates. Regarding mTORC1 hyperactivation, TSC1mKO mice significantly reduce hanging time in agreement with the previous report using in vitro force measurements (Castets 2013 doi: 10.1016/j.cmet.2013.03.015). Moreover, we found that only 4EBP1 activation, not S6K1 inactivation, improves hanging time in the TSC1mKO mouse background. The muscle function of 4EBP1mt-muscle has been analyzed in our previous report (Tsai 2015 doi: 10.1172/JCI77361).

Reviewer #2 (Remarks to the Author):

In general, the authors have done a fine job in addressing concerns expressed with initial iteration of this paper. Some concerns remain about misunderstanding about fragmentation necessarily indicating damage to NMJ and rather just as an indicator of normal NMJ remodeling. But besides that, fine job in answering my earlier concerns.

Thanks for your recognition of our previous revised work. We agree with you that we could not be certain about damaged NMJ or normal NMJ remodeling in 4EBP1mt-muscle mice. Therefore, in the discussion, we proposed that dramatic cycles of degeneration and regeneration of myofibers and NMJ contributed to NMJ fragmentation.

“The fragmentation phenotype driven by chronic 4EBP1 activation is likely a result of cycles of degeneration and regeneration of the myofibers and NMJs.”

REVIEWER COMMENTS

Reviewer #1 (Remarks to the Author):

all points were addressed

We thank reviewers for their comments. We have revised our manuscript according to the comments as indicated below.

Responses to individual reviewers:

REVIEWER COMMENTS

Reviewer #1 (Remarks to the Author):

all points were addressed

Thanks for your recognition of our previous revised work.